# Coherent State Control to Recover Quantum Entanglement and Coherence

**DOI:** 10.3390/e21100917

**Published:** 2019-09-20

**Authors:** Li-Tuo Shen, Zhi-Cheng Shi, Zhen-Biao Yang

**Affiliations:** Fujian Key Laboratory of Quantum Information and Quantum Optics, College of Physics and Information Engineering, Fuzhou University, Fuzhou 350116, China; szc2014@yeah.net

**Keywords:** coherent state, Jaynes–Cummings models, entanglement, coherence, open quantum system

## Abstract

How to analytically deal with the entanglement and coherence dynamics of separated Jaynes–Cummings nodes with continuous-variable fields is still an open question. We here generalize this model to a more common situation including either a small or large qubit-field detuning, and obtain two new analytical formulas. The X-state simplification, Fock-state shortcut and detuning-limit approximation work together in an amazingly accurate way, which agrees with the numerical results. The new formulas almost perfectly predict the two-qubit entanglement dynamics both in sudden death and rebirth phenomenon for detuning interactions. We find that when both the qubit-field detuning and amplitude of coherent states are large enough, the maximal entanglement and coherence peaks can be fully and periodically retrieved, and their revival periods both increase linearly with the increasing detuning.

## 1. Introduction

Qubit entanglement and coherence preservation are core issues in the fundamental theory and experiment of quantum optics and quantum information [1,2,3,4,5,6,7,8,9,10,11,12,13,14,15,16,17,18,19,20,21,22,23,24]. Reliable operations in quantum information processing should rely on the coherent manipulation of which information is processed or transmitted [25,26]. However, due to decoherence, where the unavoidable coupling between the real quantum system and its surrounding environment chaotically changes the target quantum state and finally disappearances of entanglement and coherence occur, quantum entanglement and coherence become so fragile and consequently go through an asymptotic decay or a sudden death [27,28,29,30].

Previous studies [5,14,20,29] have shown that the entanglement sudden death and rebirth appear in two separate Jaynes–Cummings nodes where two initial fields are both in the vacuum states, which are tough to generate and preserve due to decoherence in real experiments. The node represents the quantum subsystem of interest, i.e., a qubit and a local quantum field. Furthermore, the so-called nodes should be initially entangled in order to observe death and rebirth (revivals) of entanglement. We focus on qubit entanglement in this paper. Therefore, it is significantly important to look for powerful field resources that can lead to the long-time generation and preservation of qubit entanglement and coherence.

One common continuous-variable resource is coherent state, which contains infinite eigenstate spectrums and can be easily controlled by a classical monochromatic current in real experiments [31,32]. However, although it can be solved directly through numerical diagonalization in a truncated Hilbert space, it is still difficult to obtain the analytical time-dependent dynamics when coupled to qubits due to the complexity of infinite-dimensional Hilbert space. As far as we know, how to analytically deal with the general entanglement and coherence dynamics of separated Jaynes–Cummings nodes with coherent-state fields is still an open question [33,34,35,36,37,38], and few analytical methods can be directly used to explain their entanglement and coherence dynamics.

It has been theoretically reported [35] that one-ebit entanglement reciprocation between qubits and coherent-state fields is workable by postselection, where this postselection needs an extra step to project a mixed state into a pure state. Recent works [37,38] use an analytically novel method to prove that, even when the amplitudes of coherent-state fields are both large enough, the qubit entanglement dynamics of two resonant Jaynes–Cummings nodes can be simply explained by an exponentially decaying formula. However, due to analytical diagonalization obstacle in the infinite-dimension Hilbert space, the above analytical formula fails completely when the qubit-field interaction is not resonant, and whether the coherence and one-ebit entanglement under the detuning interaction can be fully revived is unknown yet .

To answer the above question, we here focus on generalizing the saddle point method in [38] to a more common situation including either a small or large qubit-field detuning, and obtain two new formulas analytically describing the qubit entanglement dynamics. The new formulas well explain the two-qubit entanglement dynamics both in the sudden death and in the rebirth phenomenon when the qubit-field interaction is not resonant. Especially, we find that when both the detuning and amplitude of coherent states are large enough, the maximal entanglement and coherence peaks can be fully and periodically retrieved, and their revival periods both increase linearly with the increasing detuning. When qubit-field detuning is small enough, the two-qubit entanglement exhibits the sudden death and rebirth phenomenon, and its revival peaks increase quadratically with the increasing detuning, but the revival period is delayed with a quantity quadratically depending on the detuning, while its coherence quickly oscillates and exponentially decays without sudden death and rebirth. Finally, the effect of dissipation factors on the qubit entanglement is considered.

## 2. Hamiltonian System

As shown in Figure 1, the system is described by the double Jaynes–Cummings Hamiltonian (ℏ=1)
(1)$$\begin{array}{ccc} H & = & \frac{\omega_{0}}{2}\sigma_{z}^{A}+\omega a^{†}a+G\left(\sigma_{+}^{A}a+\sigma_{-}^{A}a^{†}\right)+\frac{\omega_{0}}{2}\sigma_{z}^{B}+\omega b^{†}b+G\left(\sigma_{+}^{B}b+\sigma_{-}^{B}b^{†}\right), \end{array}$$
where $\omega_{0}$ is the transition frequency between the high level $|e_{x}\rangle$ and low level $|g_{x}\rangle$ of the qubit *x* (x=A,B). $\sigma_{z}^{x}$ and $\sigma_{\pm }^{x}$ are Pauli matrices of the qubit *x*. $a^{†}$ (*a*) and $b^{†}$ (*b*) are the creation (annihilation) operators for two single-mode fields with angular frequency ω, respectively. *G* is the qubit-field coupling strength. Here the assumption $\omega_{0}\neq \omega$, referred to as the system allowing nonzero qubit-field detuning, represents a clear distinction from the previous work with resonant coupling [38]. For simplicity, we define the detuning $\Delta =\omega_{0}-\omega$ and transform the original Hamiltonian *H* into $H_{I}$ under the interaction picture as
(2)$$\begin{array}{ccc} H_{I} & = & \frac{\Delta }{2}\left(\sigma_{z}^{A}+\sigma_{z}^{B}\right)+G\left(\sigma_{+}^{A}a+\sigma_{-}^{A}a^{†}+\sigma_{+}^{B}b+\sigma_{-}^{B}b^{†}\right), \end{array}$$
where $H_{0}=\frac{\omega }{2}\left(\sigma_{z}^{A}+\sigma_{z}^{B}\right)+\omega a^{†}a+\omega b^{†}b$ and $H_{I}=e^{iH_{0}t/\hbar }\left(H-H_{0}\right)e^{-iH_{0}t/\hbar }$ (ℏ=1). We adopt Wootters concurrence C [39] as the two-qubit entanglement measure. In order to derive new approximate formulas for the detuning situatuion, we start to analyze two-qubit entanglement dynamics under a special situation, where the field modes are initially in their coherent states with zero amplitude, i.e., vacuum-state fields.

### 2.1. Vacuum-State Fields

When the fields are initially in their vacuum states, the corresponding concurrence is
(3)$$\begin{array}{c} C=|1-\frac{2G^{2}\left[1-\cos\left(\sqrt{\Delta^{2}+4G^{2}}t\right)\right]}{\Delta^{2}+4G^{2}}|, \end{array}$$
and its oscillation period is
(4)$$\begin{array}{c} T_{v}=\frac{2k\pi }{\sqrt{\Delta^{2}+4G^{2}}},\left(k\in N\right). \end{array}$$

Compared with Equation (Equation 27) of [38], the oscillation period of the expression *C* here has an approximately inverse relation with Δ and becomes much smaller than that in Equation (Equation 27) of [38]. Figure 2a shows that the two-qubit entanglement keeps close to the maximum value C=1 for large detunings. Figure 2b shows that the period depends quadratically on the detuning, indicating that the entanglement oscillates more quickly when the detuning increases. These behaviors are very different from the system with resonant couplings [29] where the concurrence exhibits a standard Rabi oscillation with a fixed period. This is because detuning reduces the energy-exchange probability between the qubit and its local photon field, changing the period and amplitude of the Rabi oscillation. As the detuning increases, the energy coupling between entangled qubits and their respective vacuum states is very week, and the excitation energy mainly keeps in two qubits as the evolution time increases, ensuring two qubits are always in the originally entangled state.

To provide examples in which the effect of the Stark shift can be clearly identified, the dynamics of the concurrence for increasing the average photon number $\bar{n}$ is plotted in Figure 2c, where $\bar{n}$ is starting from 0 (vacuum). When $\bar{n}>0$, the Stark shift can be clearly identified that the concurrence can not remain at 1 and becomes smaller as the evolution time is longer.

To explain the above results, it is necessary to make a further simplification for the analytical concurrence in the limit of small or large detuning. In the small-detuning limit, $\sqrt{\Delta^{2}+4G^{2}}≃2G$, leading to
(5)$$\begin{array}{ccc} C & ≃ & \left|\frac{1}{2}\cos\left(2Gt\right)+\frac{2G^{2}}{\Delta^{2}+4G^{2}}\right| \end{array}$$
and
(6)$$\begin{array}{ccc} T_{v} & ≃ & \frac{k\pi }{G}\left(1-\frac{\Delta^{2}}{8G^{2}}\right), \end{array}$$
which demonstrates that both the minimum value and period of concurrence are quadratically dependent on Δ. In the large-detuning limit, $\Delta^{2}+4G^{2}≃\Delta^{2}$, leading to C≃1 and $T_{v}≃\frac{2k\pi }{\Delta }$, which demonstrates that the concurrence keeps close to 1 and the period becomes reversely proportional to Δ, explaining the fast-oscillation behavior in the concurrence. For the two-qubit coherence, its analytical result is the same to concurrence, and the effect of dissipation will exponentially reduce the two-qubit entanglement, i.e., when α=0 as proved in Equation (Equation 47) later.

In the followings, we focus on the system including two initial coherent-state fields with large amplitudes.

### 2.2. Coherent-State Fields

Assume the initial state of system to be
(7)$$\begin{array}{c} |\Psi \left(0\right)\rangle =\frac{1}{\sqrt{2}}(|e,g;\alpha ,\alpha \rangle +|g,e;\alpha ,\alpha \rangle ), \end{array}$$
where the first qubit and the first field state are in order of listing, and the coherent state is expanded by the Fock states
(8)$$\begin{array}{c} |\alpha \rangle =\sum_{n=0}^{\infty }\frac{e^{-{\left|\alpha \right|}^{2}/2}\alpha^{n}}{\sqrt{n!}}|n\rangle =\sum_{n=0}^{\infty }A_{n}|n\rangle . \end{array}$$
Therefore, the evolution dynamics become
(9)$$\begin{array}{ccc} |\Psi (t)\rangle & = & \frac{e^{-iH_{I}t}}{\sqrt{2}}(|e,g;\alpha ,\alpha \rangle +\left|g,e;\alpha ,\alpha \rangle \right)=\frac{1}{\sqrt{2}}\sum_{n=0}^{\infty }\sum_{m=0}^{\infty }A_{n}A_{m}K_{nm}, \end{array}$$
where
(10)$$\begin{array}{ccc} K_{nm} & = & \frac{G\sqrt{m}}{\gamma_{m}}\left(-iC_{n+1}+\frac{\Delta }{2\gamma_{n+1}}S_{n+1}\right)S_{m}|e,e;n,m-1\rangle \\ \; & \; & +(C_{n+1}C_{m}-\frac{i\Delta }{2\gamma_{m}}C_{n+1}S_{m}+\frac{i\Delta }{2\gamma_{n+1}}S_{n+1}C_{m}+\frac{\Delta^{2}}{4\gamma_{n+1}\gamma_{m}}S_{n+1}S_{m})|e,g;n,m\rangle \\ \; & \; & -\frac{G^{2}\sqrt{n+1}\sqrt{m}}{\gamma_{n+1}\gamma_{m}}S_{n+1}S_{m}|g,e;n+1,m-1\rangle -\frac{G\sqrt{n+1}}{\gamma_{n+1}}S_{n+1}\left(iC_{m}+\frac{\Delta }{2\gamma_{m}}S_{m}\right)|g,g;n+1,m\rangle \\ \; & \; & +\frac{G\sqrt{n}}{\gamma_{n}}S_{n}\left(-iC_{m+1}+\frac{\Delta }{2\gamma_{m+1}}S_{m+1}\right)|e,e;n-1,m\rangle -\frac{G^{2}\sqrt{n}\sqrt{m+1}}{\gamma_{n}\gamma_{m+1}}S_{n}S_{m+1}|e,g;n-1,m+1\rangle \\ \; & \; & +(C_{n}C_{m+1}+\frac{i\Delta }{2\gamma_{m+1}}C_{n}S_{m+1}-\frac{i\Delta }{2\gamma_{n}}S_{n}C_{m+1}+\frac{\Delta^{2}}{4\gamma_{n}\gamma_{m+1}}S_{n}S_{m+1})|g,e;n,m\rangle \\ \; & \; & -\frac{G\sqrt{m+1}}{\gamma_{m+1}}\left(iC_{n}+\frac{\Delta }{2\gamma_{n}}S_{n}\right)S_{m+1}|g,g;n,m+1\rangle , \end{array}$$
in which $C_{n}=\cos\left(\gamma_{n}t\right)$ and $S_{n}=\sin\left(\gamma_{n}t\right)$. Since there is detuning in the infinite-dimension Hilbert space, the joint qubit-field dynamics in Equation (Equation 9) is extremely complicated and the generally analytical solution of concurrence is hard to obtain.

Our target here is to find new analytical formulas for determining the entanglement dynamics with highly excited (nearly classical) coherent-state fields. Under the limit of small or large detuning, the new formulas are intuitively drastic simplifications for infinite dimensions in the coherent-state fields and can explain new physics features which do not appear in the situation with resonant coupling.

Based on the idea of the Fock-state shortcut [38], when the average photon number $\bar{n}$ of coherent-state fields satisfies $\bar{n}>>1$, it is feasible to replace |α〉 by $|\bar{n}\rangle$, assuming that the photon number in each coherent state obeys the Poisson distribution and centers tightly around $\bar{n}$. Note that the qubit excitation (deexcitation) transition accompanies with the absorption (emission) of one photon, the initial field state |α〉⊗|α〉 can be equivalent to the single-product Fock state $|\bar{n}\rangle ⊗|\bar{n}\rangle$ and the photon number in each field mode can be $\bar{n}$ or $\bar{n}\pm 1$ during the resonant Jaynes–Cummings interaction. Consider the coherent oscillation between |e,n> and |g,n+1>, many other quantum states with different photon numbers are truncated in the following results. Actually, these results would only be correct when the energies of the two states are close to each other and those of others are not. If it is not the case, additional quantum states could be involved in the oscillation and the following simple results would break down. However, the detuning interaction causes a virtual energy exchange between the qubit and field mode, which enhances the validity of the above assumption, meaning that the photon number centers more tightly around $\bar{n}$ in the detuning situation than that in the resonant situation.

By tracing out two field modes from Equation (Equation 9), we obtain the approximate *X*-form reduced density matrix of two qubits ρ within Γ={|e,e>,|e,g>,|g,e>,|g,g>} as follows
(11)$$\begin{array}{ccc} \rho & = & \left(\begin{array}{cccc} \rho_{11} & \rho_{x} & \rho_{x} & \rho_{x}\\ \rho_{x} & \rho_{22} & \rho_{23} & \rho_{x}\\ \rho_{x} & \rho_{23}^{*} & \rho_{33} & \rho_{x}\\ \rho_{x} & \rho_{x} & \rho_{x} & \rho_{44} \end{array}\right)\approx \left(\begin{array}{cccc} \rho_{11} & 0 & 0 & 0\\ 0 & \rho_{22} & \rho_{23} & 0\\ 0 & \rho_{23}^{*} & \rho_{33} & 0\\ 0 & 0 & 0 & \rho_{44} \end{array}\right), \end{array}$$
where $\rho_{23}^{*}$ is the conjugate complex of $\rho_{23}$ and the other small-quantity elements denoted by $\rho_{x}$ are omitted through the equal-$\bar{n}$ approximation. Thus, the concurrence has a simple form
(12)$$\begin{array}{ccc} C & = & 2\max\{0,|\rho_{23}|-\sqrt{\rho_{11}\rho_{44}}\}. \end{array}$$

Joint control of the coherent states and detuning will cause the time-dependent variation of matrix elements $\rho_{11}$, $\rho_{23}$ and $\rho_{44}$, leading to the growth or decline of two-qubit entanglement, which allows an extra freedom of detuning for controlling the entanglement than the resonant situation. Since the reduced density matrix in Equation (Equation 11) for two-qubit entanglement has been obtained by the method of Fock-state shortcut, we avoid using this method again and introduce another approximation method to obtain new analytic formulas for $\rho_{11}$, $\rho_{23}$ and $\rho_{44}$ in the limit of small or large detuning.

It is workable to give out the fully analytical expressions of the elements in Equation (Equation 11) through the tracing operation $\rho ={\mathrm{Tr}}_{a,b}[|\Psi \left(t\right)\rangle \langle \Psi \left(t\right)|]$. The result shows that the doubly infinite summations for $\rho_{23}$, $\rho_{11}$ and $\rho_{44}$ are given as
(13)$$\begin{array}{ccc} z & = & \rho_{23}=\frac{1}{2}\sum_{n=0}^{\infty }\sum_{m=0}^{\infty }[A_{n}^{2}A_{m}^{2}C_{n}C_{n+1}C_{m}C_{m+1}\\ \; & \; & +\frac{\Delta^{2}}{2\gamma_{n}\gamma_{m+1}}A_{n}^{2}A_{m}^{2}S_{n}C_{n+1}C_{m}S_{m+1}\\ \; & \; & -\frac{\Delta^{2}}{2\gamma_{m}\gamma_{m+1}}A_{n}^{2}A_{m}^{2}C_{n}C_{n+1}S_{m}S_{m+1}\\ \; & \; & +\frac{\Delta^{2}}{4\gamma_{m}\gamma_{n}}A_{n}^{2}A_{m}^{2}S_{n}C_{n+1}S_{m}C_{m+1}\\ \; & \; & +\frac{\Delta^{2}}{4\gamma_{n+1}\gamma_{m+1}}A_{n}^{2}A_{m}^{2}C_{n}S_{n+1}C_{m}S_{m+1}\\ \; & \; & +\frac{\Delta^{4}}{16\gamma_{n+1}\gamma_{n}\gamma_{m+1}\gamma_{m}}A_{n}^{2}A_{m}^{2}S_{n}S_{n+1}S_{m}S_{m+1}\\ \; & \; & -\frac{2\sqrt{n(m+1)}G^{2}}{\gamma_{n}\gamma_{m+1}}A_{n}A_{n-1}A_{m}A_{m+1}S_{n}C_{n+1}C_{m}S_{m+1}\\ \; & \; & -\frac{\sqrt{n(m+1)}\Delta^{2}G^{2}}{2\gamma_{n+1}\gamma_{n}\gamma_{m+1}\gamma_{m}}A_{n}A_{n-1}A_{m}A_{m+1}S_{n}S_{n+1}S_{m}S_{m+1}\\ \; & \; & +\frac{\sqrt{n(n-1)(m+1)(m+2)}G^{4}}{\gamma_{n}\gamma_{n-1}\gamma_{m+1}\gamma_{m+2}}A_{n}A_{n-2}A_{m}A_{m+2}\\ \; & \; & \times S_{n-1}S_{n}S_{m+1}S_{m+2}], \end{array}$$
(14)$$\begin{array}{ccc} a & = & \rho_{11}=\sum_{n=0}^{\infty }\sum_{m=0}^{\infty }[\frac{mG^{2}}{\gamma_{m}^{2}}A_{n}^{2}A_{m}^{2}C_{n+1}^{2}S_{m}^{2}\\ \; & \; & +\frac{n\Delta^{2}G^{2}}{4\gamma_{n}^{2}\gamma_{m+1}^{2}}A_{n}^{2}A_{m}^{2}S_{n}^{2}S_{m+1}^{2}\\ \; & \; & +\frac{\sqrt{(n+1)m}G^{2}}{\gamma_{n+1}\gamma_{m}}A_{n}A_{n+1}A_{m}A_{m-1}C_{n+1}S_{n+1}C_{m}S_{m}\\ \; & \; & +\frac{\sqrt{(n+1)m}\Delta^{2}G^{2}}{4\gamma_{n+1}^{2}\gamma_{m}^{2}}A_{n}A_{n+1}A_{m}A_{m-1}S_{n+1}^{2}S_{m}^{2}], \end{array}$$
and
(15)$$\begin{array}{ccc} d & = & \rho_{44}=\sum_{n=0}^{\infty }\sum_{m=0}^{\infty }[\frac{\left(n+1\right)G^{2}}{\gamma_{n+1}^{2}}A_{n}^{2}A_{m}^{2}S_{n+1}^{2}C_{m}^{2}\\ \; & \; & +\frac{\left(m+1\right)\Delta^{2}G^{2}}{4\gamma_{n}^{2}\gamma_{m+1}^{2}}A_{n}^{2}A_{m}^{2}S_{n}^{2}S_{m+1}^{2}\\ \; & \; & +\frac{\sqrt{(n+1)m}G^{2}}{\gamma_{n+1}\gamma_{m}}A_{n}A_{n+1}A_{m}A_{m-1}C_{n+1}S_{n+1}C_{m}S_{m}\\ \; & \; & +\frac{\sqrt{(n+1)m}\Delta^{2}G^{2}}{4\gamma_{n+1}^{2}\gamma_{m}^{2}}A_{n}A_{n+1}A_{m}A_{m-1}S_{n+1}^{2}S_{m}^{2}], \end{array}$$
respectively. These summations indicate that the qubits couple to an unclosed space of infinite states and the detuning mainly causes a nonlinear effect during the coupling process. It is not possible to complete these summations under general conditions, but their analytical solutions can be found when the coherent states are nearly classical under two limits, i.e., the limit of small or large detuning.

To approximate infinite summations into integrals, the Stirling equation is used to replace the term n! as follows
(16)$$\begin{array}{c} n!=\sqrt{2\pi n}n^{n}e^{-n}. \end{array}$$
When $\bar{n}≃\alpha^{2}\gg 1$, it is feasible to introduce an error-deviation order of $1/\bar{n}$ centering near the Poisson peak $n\approx m=\bar{n}$ and the terms $A_{n\pm 1}A_{n}\approx A_{n\pm 2}A_{n}\approx A_{n}^{2}$. Thus we obtain the simplification form for infinite summations
(17)$$\begin{array}{ccc} z & = & \frac{1}{2}[{\left(\sum_{n=0}^{\infty }A_{n}^{2}C_{n}C_{n+1}\right)}^{2}-\frac{\Delta^{2}G^{2}}{2}{\left(\sum_{n=0}^{\infty }\frac{\sqrt{n}}{\gamma_{n}^{2}}A_{n}^{2}S_{n}S_{n+1}\right)}^{2}\\ \; & \; & +\frac{\Delta^{2}}{2}\left(\sum_{n=0}^{\infty }\frac{A_{n}^{2}}{\gamma_{n}}S_{n}C_{n+1}\right)\left(\sum_{m=0}^{\infty }\frac{A_{m}^{2}}{\gamma_{m}}C_{m}S_{m+1}\right)\\ \; & \; & -2G^{2}\left(\sum_{n=0}^{\infty }\frac{\sqrt{n}}{\gamma_{n}}A_{n}^{2}S_{n}C_{n+1}\right)\left(\sum_{m=0}^{\infty }\frac{\sqrt{m}}{\gamma_{m}}A_{m}^{2}C_{m}S_{m+1}\right)\\ \; & \; & -\frac{\Delta^{2}}{2}\left(\sum_{n=0}^{\infty }\frac{A_{n}^{2}}{\gamma_{n}}C_{n}C_{n+1}\right)\left(\sum_{m=0}^{\infty }\frac{A_{m}^{2}}{\gamma_{m}}S_{m}S_{m+1}\right)\\ \; & \; & +\frac{\Delta^{2}}{4}{\left(\sum_{n=0}^{\infty }\frac{A_{n}^{2}}{\gamma_{n}}S_{n}C_{n+1}\right)}^{2}+\frac{\Delta^{2}}{4}{\left(\sum_{n=0}^{\infty }\frac{A_{n}^{2}}{\gamma_{n}}C_{n}S_{n+1}\right)}^{2}\\ \; & \; & +\frac{\Delta^{4}}{16}{\left(\sum_{n=0}^{\infty }\frac{A_{n}^{2}}{\gamma_{n}^{2}}S_{n}S_{n+1}\right)}^{2}+G^{4}{\left(\sum_{n=0}^{\infty }\frac{nA_{n}^{2}}{\gamma_{n}^{2}}S_{n}S_{n+1}\right)}^{2}] \end{array}$$
and
(18)$$\begin{array}{ccc} a & \approx & d=G^{2}[\left(\sum_{n=0}^{\infty }\frac{\sqrt{n}}{\gamma_{n}}A_{n}^{2}C_{n}C_{n}\right)\left(\sum_{n=0}^{\infty }\frac{\sqrt{n}}{\gamma_{n}}A_{n}^{2}S_{n}S_{n}\right)\\ \; & \; & +{\left(\sum_{n=0}^{\infty }\frac{\sqrt{n}}{\gamma_{n}}A_{n}^{2}C_{n}S_{n}\right)}^{2}+\frac{\Delta^{2}}{2}{\left(\sum_{n=0}^{\infty }\frac{\sqrt{n}}{\gamma_{n}^{2}}A_{n}^{2}S_{n}S_{n}\right)}^{2}]. \end{array}$$

Note that $\rho_{11}\approx \rho_{44}$ in Equation (Equation 18) is valid for any detuning. To further simplify the above summations, we rewrite $C_{n}C_{n+1}$ as
(19)$$\begin{array}{ccc} C_{n}C_{n+1} & = & \frac{1}{2}\left\{\cos\left[\left(\gamma_{n}+\gamma_{n+1}\right)t\right]+\cos\left[\left(\gamma_{n}-\gamma_{n+1}\right)t\right]\right\}, \end{array}$$
and the approximation of large $\bar{n}$ is used for expanding the term $\gamma_{n+1}$ in Equation (Equation 19)
(20)$$\begin{array}{c} \gamma_{n+1}≃\gamma_{n}+\frac{G^{2}}{2\gamma_{n}}, \end{array}$$
which transforms $C_{n}C_{n+1}$ into
(21)$$\begin{array}{c} C_{n}C_{n+1}≃\frac{1}{2}\left[\cos\left(\frac{G^{2}t}{2\gamma_{n}}\right)+\cos\left(2\gamma_{n}t\right)\right]. \end{array}$$

Similarly, the other useful approximations are
(22)$$\begin{array}{c} S_{n}S_{n+1}≃\frac{1}{2}\left[\cos\left(\frac{G^{2}t}{2\gamma_{n}}\right)-\cos\left(2\gamma_{n}t\right)\right], \end{array}$$
(23)$$\begin{array}{c} C_{n}S_{n+1}≃\frac{1}{2}\left[\sin\left(2\gamma_{n}t\right)+\sin\left(\frac{G^{2}t}{2\gamma_{n}}\right)\right], \end{array}$$
(24)$$\begin{array}{c} S_{n}C_{n+1}≃\frac{1}{2}\left[\sin\left(2\gamma_{n}t\right)-\sin\left(\frac{G^{2}t}{2\gamma_{n}}\right)\right]. \end{array}$$

With these approximations and the identities
(25)$$\begin{array}{c} C_{n}^{2}=\frac{1+\cos(2\gamma_{n}t)}{2} \end{array}$$
and
(26)$$\begin{array}{c} S_{n}^{2}=\frac{1-\cos(2\gamma_{n}t)}{2}, \end{array}$$
we can simplify the concurrence further as
(27)$$\begin{array}{ccc} z & - & \sqrt{ad}=\frac{1}{8}\{{\left[\sum_{n=0}^{\infty }A_{n}^{2}\cos\left(2\gamma_{n}t\right)\right]}^{2}+{\left[\sum_{n=0}^{\infty }A_{n}^{2}\cos\left(\frac{G^{2}t}{2\gamma_{n}}\right)\right]}^{2}\\ \; & \; & +2\left[\sum_{n=0}^{\infty }A_{n}^{2}\cos\left(2\gamma_{n}t\right)\right]\left[\sum_{n=0}^{\infty }A_{n}^{2}\cos\left(\frac{G^{2}t}{2\gamma_{n}}\right)\right]\}\\ \; & \; & +\frac{\Delta^{2}}{8}{\left[\sum_{n=0}^{\infty }\frac{A_{n}^{2}}{\gamma_{n}}\sin\left(2\gamma_{n}t\right)\right]}^{2}-\frac{\Delta^{2}}{16}\{{\left[\sum_{n=0}^{\infty }\frac{A_{n}^{2}}{\gamma_{n}}\cos\left(\frac{G^{2}t}{2\gamma_{n}}\right)\right]}^{2}\\ \; & \; & -{\left[\sum_{n=0}^{\infty }\frac{A_{n}^{2}}{\gamma_{n}}\cos\left(2\gamma_{n}t\right)\right]}^{2}\}-\frac{G^{2}}{4}\{2{\left[\sum_{n=0}^{\infty }\frac{\sqrt{n}A_{n}^{2}}{\gamma_{n}}\sin\left(2\gamma_{n}t\right)\right]}^{2}\\ \; & \; & -{\left[\sum_{n=0}^{\infty }\frac{\sqrt{n}A_{n}^{2}}{\gamma_{n}}\sin\left(\frac{G^{2}t}{2\gamma_{n}}\right)\right]}^{2}\}-\frac{G^{2}}{4}\{{\left(\sum_{n=0}^{\infty }\frac{\sqrt{n}A_{n}^{2}}{\gamma_{n}}\right)}^{2}\\ \; & \; & -{\left[\sum_{n=0}^{\infty }\frac{\sqrt{n}A_{n}^{2}}{\gamma_{n}}\cos\left(2\gamma_{n}t\right)\right]}^{2}\}+\frac{\Delta^{4}}{128}\{{\left[\sum_{n=0}^{\infty }\frac{A_{n}^{2}}{\gamma_{n}^{2}}\cos\left(\frac{G^{2}t}{2\gamma_{n}}\right)\right]}^{2}\\ \; & \; & -2\left[\sum_{n=0}^{\infty }\frac{A_{n}^{2}}{\gamma_{n}^{2}}\cos\left(2\gamma_{n}t\right)\right]\left[\sum_{n=0}^{\infty }\frac{A_{n}^{2}}{\gamma_{n}^{2}}\cos\left(\frac{G^{2}t}{2\gamma_{n}}\right)\right]\\ \; & \; & +{\left[\sum_{n=0}^{\infty }\frac{A_{n}^{2}}{\gamma_{n}^{2}}\cos\left(2\gamma_{n}t\right)\right]}^{2}\}-\frac{\Delta^{2}G^{2}}{16}\{{\left[\sum_{n=0}^{\infty }\frac{\sqrt{n}A_{n}^{2}}{\gamma_{n}^{2}}\cos\left(\frac{G^{2}t}{2\gamma_{n}}\right)\right]}^{2}\\ \; & \; & -2\left[\sum_{n=0}^{\infty }\frac{\sqrt{n}A_{n}^{2}}{\gamma_{n}^{2}}\cos\left(2\gamma_{n}t\right)\right]\left[\sum_{n=0}^{\infty }\frac{\sqrt{n}A_{n}^{2}}{\gamma_{n}^{2}}\cos\left(\frac{G^{2}t}{2\gamma_{n}}\right)\right]\\ \; & \; & +{\left[\sum_{n=0}^{\infty }\frac{\sqrt{n}A_{n}^{2}}{\gamma_{n}^{2}}\cos\left(2\gamma_{n}t\right)\right]}^{2}\}-\frac{\Delta^{2}G^{2}}{8}\{{\left(\sum_{n=0}^{\infty }\frac{\sqrt{n}A_{n}^{2}}{\gamma_{n}^{2}}\right)}^{2}\\ \; & \; & -2\left(\sum_{n=0}^{\infty }\frac{\sqrt{n}A_{n}^{2}}{\gamma_{n}^{2}}\right)\left[\sum_{n=0}^{\infty }\frac{\sqrt{n}A_{n}^{2}}{\gamma_{n}^{2}}\cos\left(2\gamma_{n}t\right)\right]\\ \; & \; & +{\left[\sum_{n=0}^{\infty }\frac{\sqrt{n}A_{n}^{2}}{\gamma_{n}^{2}}\cos\left(2\gamma_{n}t\right)\right]}^{2}\}+\frac{G^{4}}{8}\{{\left[\frac{nA_{n}^{2}}{\gamma_{n}^{2}}\cos\left(\frac{G^{2}t}{2\gamma_{n}}\right)\right]}^{2}\\ \; & \; & -2\left[\sum_{n=0}^{\infty }\frac{nA_{n}^{2}}{\gamma_{n}^{2}}\cos\left(2\gamma_{n}t\right)\right]\left[\sum_{n=0}^{\infty }\frac{nA_{n}^{2}}{\gamma_{n}^{2}}\cos\left(\frac{G^{2}t}{2\gamma_{n}}\right)\right]\\ \; & \; & +{\left[\sum_{n=0}^{\infty }\frac{nA_{n}^{2}}{\gamma_{n}^{2}}\cos\left(2\gamma_{n}t\right)\right]}^{2}\}. \end{array}$$

Since the detuning appears in the square root of the Lambert W function [see Appendix A], it is impossible to fully calculate analytical results for the summations in Equation (Equation 27) under a general detuning by the saddle point method. However, we find that for the limit of small or large detuning, it is feasible to calculate analytically these summations through writing them as integrals, where the discrete integer *n* is treated as continuous when $\bar{n}$ is large enough.

In the following section, we focus on exploring the concurrence in two limits of detuning that can be analytically treated by the saddle point method.

#### 2.2.1. Small-Detuning Limit

The first extreme situation we focus on is the small-detuning limit, i.e., $\Delta <<2G\sqrt{\bar{n}}$. Under this limit, the summations in Equation (Equation 27) can be simplified further as
(28)$$\begin{array}{ccc} \left|z\right| & - & \sqrt{ad}\approx \frac{1}{4}\{{\left[\sum_{n=0}^{\infty }A_{n}^{2}\cos\left(\frac{G^{2}t}{2\gamma_{n}}\right)\right]}^{2}+{\left[\sum_{n=0}^{\infty }A_{n}^{2}\sin\left(\frac{G^{2}t}{2\gamma_{n}}\right)\right]}^{2}\\ \; & \; & +2{\left[\sum_{n=0}^{\infty }A_{n}^{2}\cos\left(2\gamma_{n}t\right)\right]}^{2}-2{\left[\sum_{n=0}^{\infty }A_{n}^{2}\sin\left(2\gamma_{n}t\right)\right]}^{2}-1\}, \end{array}$$
where $\sqrt{n}G≃\gamma_{n}$ is used and high-order terms than $O(\frac{1}{\gamma_{n}})$ have been omitted. Therefore, it needs to calculate four integrals
(29)$$\begin{array}{c} I_{1}=\int_{0}^{\infty }A_{n}^{2}\cos\left(\frac{G^{2}t}{\sqrt{\Delta^{2}+4nG^{2}}}\right)dn, \end{array}$$
(30)$$\begin{array}{c} I_{2}=\int_{0}^{\infty }A_{n}^{2}\sin\left(\frac{G^{2}t}{\sqrt{\Delta^{2}+4nG^{2}}}\right)dn, \end{array}$$
(31)$$\begin{array}{c} I_{3}=\int_{0}^{\infty }A_{n}^{2}\cos\left(\sqrt{\Delta^{2}+4nG^{2}}t\right)dn, \end{array}$$
and
(32)$$\begin{array}{c} I_{4}=\int_{0}^{\infty }A_{n}^{2}\sin\left(\sqrt{\Delta^{2}+4nG^{2}}t\right)dn. \end{array}$$
These integrals can be combined as $I_{1}+iI_{2}=I_{12}$ and $I_{3}+iI_{4}=I_{34}$ for dealing with the exponentials $\frac{G^{2}t}{\sqrt{\Delta^{2}+4nG^{2}}}$ and $\sqrt{\Delta^{2}+4nG^{2}}t$, respectively. Based on the Stirling equation and Euler formula, the integrals can be approximated as
(33)$$\begin{array}{c} I_{12}≃\int_{0}^{\infty }e^{-\alpha^{2}}\frac{\alpha^{2n}e^{n}}{\sqrt{2\pi n}n^{n}}e^{i\frac{G^{2}t}{\sqrt{\Delta^{2}+4nG^{2}}}}dn \end{array}$$
and
(34)$$\begin{array}{c} I_{34}≃\int_{0}^{\infty }e^{-\alpha^{2}}\frac{\alpha^{2n}e^{n}}{\sqrt{2\pi n}n^{n}}e^{i\sqrt{\Delta^{2}+4nG^{2}}t}dn. \end{array}$$

It is workable to use the saddle point method [38] to analytically calculate the integrals $I_{12}$ and $I_{34}$, and details for dealing with small-detuning terms of the exponentials are contained in the Appendix A. Thus, analytical expressions for $I_{12}$ and $I_{34}$ are found to be
(35)$$\begin{array}{c} I_{12}≃e^{\tau^{2}\left(-\frac{1}{32\alpha^{4}}+\frac{3\Delta^{2}}{128\alpha^{6}G^{2}}\right)}e^{i\tau \left(\frac{1}{2\alpha }-\frac{\Delta^{2}}{16\alpha^{3}G^{2}}\right)} \end{array}$$
and
(36)$$\begin{array}{ccc} I_{34} & ≃ & e^{-\tau^{2}\left(\frac{1}{2}+\frac{\Delta^{2}}{8\alpha^{2}G^{2}}\right)}e^{i\tau \left(2\alpha +\frac{\Delta^{2}}{4\alpha G^{2}}\right)}+\sum_{k=1,2,...}\sqrt{\frac{1}{\pi k}}\\ & \times & e^{\frac{-1}{1+\pi^{2}k^{2}}\left(1+\frac{\Delta^{2}}{8\alpha^{2}G^{2}}\right){\left(\tau -2\pi k\alpha \right)}^{2}+\frac{\Delta^{2}\pi k}{4\alpha \left(1+\pi^{2}k^{2}\right)G^{2}}\left(\tau -2\pi k\alpha \right)}\\ & \times & e^{i\left\{\left[{\left(-1\right)}^{k}\frac{\Delta^{2}}{2G^{2}}-1\right]\pi k+{\left(-1\right)}^{k}\left[2\pi k\alpha^{2}+\left(2\alpha +\frac{\Delta^{2}}{4\alpha G^{2}}\right)\left(\tau -2\pi k\alpha \right)\right]\right\}}, \end{array}$$
where τ=Gt. By putting these approximation results into Equation (Equation 28), a new formula for two-qubit entanglement determiner with small detunings is obtained
(37)$$\begin{array}{ccc} |\rho_{23}| & - & \sqrt{\rho_{11}\rho_{44}}≃\frac{1}{4}\left(e^{\frac{-\tau^{2}}{16\alpha^{4}}+\frac{3\Delta^{2}\tau^{2}}{64\alpha^{6}G^{2}}}-1\right)\\ \; & \; & +\frac{1}{4}e^{\frac{-(4\alpha^{2}G^{2}+\Delta^{2})}{8\alpha^{2}G^{2}}\tau^{2}}\cos\left[\frac{\left(8\alpha^{2}G^{2}+\Delta^{2}\right)}{2\alpha G^{2}}\tau \right]\\ \; & \; & +\sum_{k=1,2,...}\frac{1}{2\pi k}e^{\frac{-[\left(8\alpha^{2}G^{2}+\Delta^{2}\right)\tau -4\left(4\alpha^{2}G^{2}+\Delta^{2}\right)\pi k\alpha ]}{4\alpha^{2}\left(1+\pi^{2}k^{2}\right)G^{2}}\left(\tau -2\pi k\alpha \right)}\\ \; & \; & \times \cos\left[\frac{\left(8\alpha^{2}G^{2}+\Delta^{2}\right)}{2\alpha G^{2}}\tau -8\pi k\alpha^{2}\right]. \end{array}$$

For this formula we have used the fact that only the term with the corresponding *k* around τ=2πkα or $\tau =4\left(4\alpha^{2}G^{2}+\Delta^{2}\right)\pi k\alpha /\left(8\alpha^{2}G^{2}+\Delta^{2}\right)$ gives a main contribution to the sums. Compared with Equation (Equation 60) of [38], this formula here in Equation (Equation 37) contains new contributions of $\Delta^{2}$ in the index of exponential function and the period of cosine function. The contribution to τ=2πkα or $\tau =4\left(4\alpha^{2}G^{2}+\Delta^{2}\right)\pi k\alpha /\left(8\alpha^{2}G^{2}+\Delta^{2}\right)$ from any other $k^{\prime }$ decays exponentially with the distance from *k*, i.e., proportional to
(38)$$\begin{array}{c} \exp\left\{\frac{-\pi^{2}\left[\left(8\alpha^{2}G^{2}+\Delta^{2}\right)k-2\left(4\alpha^{2}G^{2}+\Delta^{2}\right)k^{\prime }\right]}{\left(1+\pi^{2}k^{\prime }{}^{2}\right)G^{2}}\left(k-k^{\prime }\right)\right\}, \end{array}$$
which decays with the square of increasing detuning. This leads to the unique revival period $T_{c}=4\left(4\alpha^{2}G^{2}+\Delta^{2}\right)\pi k\alpha /\left(8\alpha^{2}G^{2}+\Delta^{2}\right)$ and the relative revival envelope height
(39)$$\begin{array}{c} \frac{1}{\pi k}-\frac{1-\exp[\left(-4\alpha^{2}G^{2}+2\Delta^{2}\right)\tau^{2}/\left(64\alpha^{6}G^{2}\right)]}{2}. \end{array}$$

Compared with Equation (Equation 61) of [38], this formula here in Equation (Equation 39) contains a new contribution $2\Delta^{2}\tau^{2}/\left(64\alpha^{2}G^{2}\right)$ in the index of exponential function, and the relative revival envelope height grows with the increasing Δ. Thus the formula in Equation (Equation 37) is the main analytical result for each step index *k* with small detuning and reduces to that with resonant coupling [38] at both the decay exponent and revival period, where resonance can be viewed as a special limit of small detuning.

To perform the numerical simulations, we use the Hamiltonian given in Equation (Equation 2) by truncating the Fock basis and calculating the reduced density matrix in each excitation subspace, then summing over all the subspaces to get the total reduced density matrix. This Fock-basis truncation is based on the precision accuracy of numerical concurrence, and the total excitation number is cut off at $2\bar{n}$. For example, when coherent states contain $\bar{n}=100$ bosons in each of the cavity modes, the total excitation number is cut off at 200. This is because when $\bar{n}>>1$, the width of the photon number distribution obeys $1<<\Delta n<<\bar{n}$. Therefore, it is safe to make this Fock-basis truncation.

In Figure 3a,b, we plot the long-time entanglement dynamics with small detunings for analytical and numerical calculations when two qubits are exposed to two coherent fields each with a large average photon number $\bar{n}=100$, respectively. In Figure 3c, $C_{f}$ value is the first concurrence revival peak. We find that the entanglement exhibits sudden death and rebirth phenomenon, and its revival peaks are not fully complete but increase quadratically with the increasing detuning, as shown in Figure 3c. The revival period is delayed with a quantity quadratically depending on the detuning, as shown in Figure 3d. The numerical results in Figure 3b are not perfectly predicted by the corresponding analytical results in Figure 3a, and their main difference is the absence of Rabi-type oscillations during the revivals, i.e., the disappearance of tiny revivals in numerical results, which are not contained in the new formula of Equation (Equation 37). This is because the entanglement between the two subsystems is conserved due to no interaction between them. The small detuning in each subsystem makes each photon field deviate from the coherent state so that an ebit for the qubits cannot be completely transferred to the fields, i.e., two qubits become a mixed state and their entanglement quickly vanishes by tracing out two fields as the evolution time increases, so that the two qubits can not be fully recovered.

#### 2.2.2. Large-Detuning Limit

The second extreme situation we focus on is the large-detuning limit, i.e., $\Delta >>2G\sqrt{\bar{n}}$. Under this limit, the summations in Equation (Equation 27) can be simplified further as
(40)$$\begin{array}{ccc} \left|z\right| & - & \sqrt{ad}\approx \frac{1}{8}\{3{\left[\sum_{n=0}^{\infty }A_{n}^{2}\cos\left(2\gamma_{n}t\right)\right]}^{2}-{\left[\sum_{n=0}^{\infty }A_{n}^{2}\cos\left(\frac{G^{2}t}{2\gamma_{n}}\right)\right]}^{2}\\ \; & \; & +2\left[\sum_{n=0}^{\infty }A_{n}^{2}\cos\left(2\gamma_{n}t\right)\right]\left[\sum_{n=0}^{\infty }A_{n}^{2}\cos\left(\frac{G^{2}t}{2\gamma_{n}}\right)\right]\}\\ \; & \; & +\frac{1}{2}{\left[\sum_{n=0}^{\infty }A_{n}^{2}\sin\left(2\gamma_{n}t\right)\right]}^{2}, \end{array}$$
where $\sqrt{\Delta^{2}+4nG^{2}}≃\Delta$ is used and higher-order terms than $O(\frac{\Delta^{2}}{\gamma_{n}^{2}})$ have been omitted. We need to recalculate the integrals in Equations (Equation 33) and (Equation 34), where large-detuning terms can be directly removed from the square root of the Lambert W function, and these derivation details are contained in the Appendix A. We obtain the analytical expressions of $I_{12}$ and $I_{34}$ as follows
(41)$$\begin{array}{c} I_{12}≃e^{-\frac{4\alpha^{2}G^{6}\tau^{2}}{\Delta^{6}}}e^{i\frac{G\tau }{\Delta }} \end{array}$$
and
(42)$$\begin{array}{c} I_{34}≃e^{-\alpha^{2}\left[1-\cos\left(\frac{2G\tau }{\Delta }\right)\right]}e^{i[\frac{\Delta \tau }{G}-\frac{G\tau }{\Delta }+\alpha^{2}\sin\left(\frac{2G\tau }{\Delta }\right)]}. \end{array}$$
With these results in Equation (Equation 40), a new formula for two-qubit entanglement determiner with large detunings is obtained
(43)$$\begin{array}{ccc} |\rho_{23}|- & \sqrt{\rho_{11}\rho_{44}} & ≃\frac{1}{2}e^{-2\alpha^{2}\left[1-\cos\left(\frac{2G\tau }{\Delta }\right)\right]}-\frac{1}{8}\{-e^{-\frac{4\alpha^{2}G^{6}\tau^{2}}{\Delta^{6}}}\\ \; & \; & \times \cos\left(\frac{G\tau }{\Delta }\right)+e^{-\alpha^{2}\left[1-\cos\left(\frac{2G\tau }{\Delta }\right)\right]}\\ \; & \; & \times \cos\left[\frac{\Delta \tau }{G}-\frac{G\tau }{\Delta }+\alpha^{2}\sin\left(\frac{2G\tau }{\Delta }\right)\right]{\}}^{2}. \end{array}$$
Here, different from the cases of resonance and small detuning, the principal exponential exhibits a periodical oscillation between the minimum $e^{-\alpha^{2}}$ and the maximum $e^{0}$ values, and does not decay monotonously with the increasing time, giving a main contribution to the sums around τ=Δkπ/G, where τ does not depend on the average photon number. It is interesting to see that around τ=Δkπ/G, the relative revival envelope height is the maximum concurrence C=1, meaning that the entanglement recovery can be fully complete without any postselection operation. This result is surprisingly different from the result with resonance or small detuning where the maximal entanglement is never fully complete.

In Figure 4a,b, we plot the long-time entanglement dynamics with large detunings for analytical and numerical calculations when two qubits are exposed to two coherent fields with a large average photon number $\bar{n}=100$, respectively. We find that the two-qubit entanglement also exhibits sudden death and rebirth phenomenon, and its revival peaks are fully complete without depending on the detuning, but the revival period increases linearly with the increasing detuning, as shown in Figure 4c,d. Although the new formula here is very different from that in the small-detuning limit, only the absence of Rabi-type oscillations during the revivals of numerical results in Figure 4b are not perfectly predicted by the corresponding analytical results in Figure 4a, which is generically similar to the revivals presented for one qubit inversion of quantum revivals with detunings [40].

The physics explanation is that under the large-detuning limit, the dispersive interaction between the qubit and field causes a Stark movement in the field frequency, which depends on the qubit state. This Stark movement leads to the opposite phase shifts between two field components, which generates entanglement between the qubit and field. When this phase difference accumulates to a certain extent, the qubit and field approximately becomes the maximally entangled state, but two qubits become a mixed state and their entanglement vanishes by tracing out two fields. However, when this phase difference is 2π, the qubit and field are not entangled and the two-qubit entanglement can be fully recovered. While in the small-detuning limit, this phase difference is impossible to achieve 2π and the two qubits can not be fully recovered. This physical process has an essential difference with that of virtual energy exchanging realized for two qubits and one vacuum field in cavity-quantum-electrodynamics system [41,42].

#### 2.2.3. Further Discussion

To see the transition from small to large detunings, we numerically simulate the concurrence dynamics for general detunings in Figure 5. From Figure 5a, it is obviously to see that each concurrence curve has a similar revival pattern and changes regularly from small to large detunings. Although the above method cannot analytically predict the concurrence dynamics for moderate detunings, it is still possible to mathematically fit the characteristics of the first revival envelope when it transits from small to large detunings. By mathematically fitting the first revival envelope in Figure 5b,c, it is interesting to find that the revival peak quadratically depends on the detuning, but the revival period linearly relates with the detuning. This result exhibits a detuning-dependence discrepancy with that under the small or large detuning.

To quantitatively show the quasiperiodic modulations by other amplitudes of the coherent state, we plot the concurrence dynamics with other average photon numbers in Figure 6. In Figure 6a,b, detuning modulation exhibits with different revival periods. For small detunings in Figure 6c, the revival period increases linearly with the increasing $\bar{n}$, i.e., $T_{f}\propto 2\sqrt{\bar{n}}\pi /G$. For large detunings in Figure 6d, the revival period has a fixed value without depending on $\bar{n}$ and the increasing $\bar{n}$ narrows the revival envelope in the time domain. Even when $\bar{n}$ decreases to 25, the analytical results well predict the numerical results. This result demonstrates that two new formulas in Equations (Equation 37) and (Equation 43) are workable for a wide range of the average photon number.

### 2.3. Quantum Coherence

Another important issue is to answer the question of whether it is possible to fully recover the coherence of two qubits from the infinite-dimension fields, i.e., coherent-state fields. To answer this question, we examine the two-qubit coherence mainly caused by detuning modulations in this section.

We first assess the quantum coherence dynamics of two qubits by computing the $l_{1}$ norm of coherence $C_{l_{1}}=\sum_{x\neq y}\left|\rho_{xy}\right|$ [43], where $\rho_{xy}$ is off-diagonal elements of density matrix ρ in the basis Γ. In general, $0\leq C_{l_{1}}\leq d-1$, where *d* is the dimension of ρ. According to Equation (Equation 11), we obtain the analytical expression of $C_{l_{1}}$, which takes a simple form $C_{l_{1}}=2\left|\rho_{23}\right|$, and d=4. The analytical formula of coherence for the small-detuning limit is ($\Delta <<2G\sqrt{\bar{n}}$)
(44)$$\begin{array}{ccc} C_{l_{1}} & ≃ & |\frac{1}{4}e^{\frac{-(4\alpha^{2}G^{2}+\Delta^{2})}{8\alpha^{2}G^{2}}\tau^{2}}\cos\left[\frac{\left(8\alpha^{2}G^{2}+\Delta^{2}\right)}{2\alpha G^{2}}\tau \right]\\ \; & \; & +\sum_{k=1,2,...}\frac{1}{2\pi k}e^{\frac{-[\left(8\alpha^{2}G^{2}+\Delta^{2}\right)\tau -4\left(4\alpha^{2}G^{2}+\Delta^{2}\right)\pi k\alpha ]}{4\alpha^{2}\left(1+\pi^{2}k^{2}\right)G^{2}}\left(\tau -2\pi k\alpha \right)}\\ \; & \; & \times \cos\left[\frac{\left(8\alpha^{2}G^{2}+\Delta^{2}\right)}{2\alpha G^{2}}\tau -8\pi k\alpha^{2}\right]+\frac{1}{2}e^{\frac{-\tau^{2}}{16\alpha^{4}}+\frac{3\Delta^{2}\tau^{2}}{64\alpha^{6}G^{2}}}|, \end{array}$$
while for the large-detuning limit is ($\Delta >>2G\sqrt{\bar{n}}$)
(45)$$\begin{array}{ccc} C_{l_{1}} & ≃ & |e^{-2\alpha^{2}\left[1-\cos\left(\frac{2G\tau }{\Delta }\right)\right]}-\frac{1}{4}\{-e^{-\frac{4\alpha^{2}G^{6}\tau^{2}}{\Delta^{6}}}\cos\left(\frac{G\tau }{\Delta }\right)\\ \; & \; & +e^{-\alpha^{2}\left[1-\cos\left(\frac{2G\tau }{\Delta }\right)\right]}\cos\left[\frac{\Delta \tau }{G}-\frac{G\tau }{\Delta }+\alpha^{2}\sin\left(\frac{2G\tau }{\Delta }\right)\right]{\}}^{2}|. \end{array}$$

In Figure 7, we numerically simulate the long-time dynamics of two-qubit coherence with small and large detunings. For the small-detuning limit, we find that the coherence increases from $C_{l_{1}}=1$ to its maximum $C_{l_{1}}=3$ around the time $\tau =2\left(4\alpha^{2}G^{2}+\Delta^{2}\right)\pi k\alpha /\left(8\alpha^{2}G^{2}+\Delta^{2}\right)$ , i.e., two qubits evolve from their maximally entangled state $\frac{1}{\sqrt{2}}(|eg\rangle +\left|ge\rangle \right)$ to maximally coherent state $\frac{1}{2}(|ee\rangle +|eg\rangle +|ge\rangle +\left|gg\rangle \right)$, then decay exponentially to zero with the increasing time. In contrast, for the large-detuning limit, we find that the coherence has an initially high value and exponentially decays to a relatively low non-zero value at the beginning. After a time period τ=Δkπ/G, the coherence is fully recovered to the initial high value, i.e., two qubits evolve to a state between the maximally entangled and maximally coherent states. This result is qualitatively different from the concurrence result. When the detuning is large, the coherence remains nonzero during the vanishing of entanglement, which does not exhibit the sudden death phenomenon. But, when the detuning is small, the coherence may inversely increase during the vanishing of entanglement.

It is of interest to see what the photon states are when the qubit states evolve to a maximal coherence and minimal coherence. In Figure 8, we plot the evolution dynamics of the average photon number and width in distribution. We find that the average photon number for each subsystem becomes maximum when the qubit states evolve to a maximal coherence, but becomes minimum when the qubit states evolve to a minimal coherence, and width in distribution has the similar evolution with that in the qubit coherence.

## 3. Effect of Dissipation

To see the effect of dissipation factors on the two-qubit entanglement, we numerically simulate the system’s master equation by the approach of quantum trajectory [44]. This approach assumes that when the system contains the rates of photon decay κ and qubit spontaneous emission η, the system evolves approximately under a non-Hermitian Hamiltonian
(46)$$\begin{array}{c} H_{\kappa ,\eta }=H_{I}-\frac{i}{2}\sum_{y=1}^{4}Y_{y}^{†}Y_{y}, \end{array}$$
where $Y_{1}=\sqrt{\kappa }a$, $Y_{2}=\sqrt{\kappa }b$, $Y_{3}=\sqrt{\eta }\sigma_{-}^{A}$, and $Y_{4}=\sqrt{\eta }\sigma_{-}^{B}$ are the collapse operators causing instantaneous quantum jumps. Equation (Equation 46) is a Hamiltonian interaction in the Schrödinger picture, where the operator is time-independent but the state vector is time-dependent.

In Figure 9, we numerically simulate the effect of dissipation factor κ or η on the two-qubit entanglement both for small and large detunings. The result shows that the entanglement decreases exponentially as κ or η increases, and is more robust against the qubit spontaneous emission than photon decay. To explain this exponential decay behavior of entanglement, it is necessary to make an assumption κ=η=λ for obtaining the analytical solution
(47)$$\begin{array}{c} C_{\kappa ,\eta }≃e^{-2\lambda \alpha^{2}t}C_{f}, \end{array}$$
where a new factor $e^{-2\lambda \alpha^{2}t}$ appears compared with the original concurrence $C_{f}$. This factor has a decreasing exponential parameter linear with dissipation rates and square with the amplitude of coherent states, which causes exponential decays in the original concurrence.

## 4. Conclusion

This paper generalizes the method reported in [38], which was restricted to the resonant situation, to a more general situation including qubit-field detunings. Based on numerical simulations and analytically new formulas, we demonstrate that the X-state simplification, Fock-state shortcut and detuning-limit approximation work together in an amazingly accurate way, which agrees with the numerical results. Although the numerical to analytic agreements are not perfect, it is safe to say that the new formulas can predict the numerical results under a wide range of average photon numbers in the coherent state. Especially, we find that when both the detuning and amplitude of coherent states are large enough, the maximal entanglement and coherence peaks can be fully and periodically retrieved and their revival periods both increase linearly with the increasing detuning. Finally, the effect of dissipation factors on the qubit entanglement is analyzed.

The work is important because the new formulas reveal the analytical relation between the entanglement evolution dynamics and detunings, which further clarifies the physics mechanism of entanglement sudden death and rebirth and provides a basic solution for direct use in any real systems. In the future, we want to further study the situation with more qubits and try to seek a general detuning solution under special continuous variables.

## Figures and Tables

**Figure 1 entropy-21-00917-f001:**
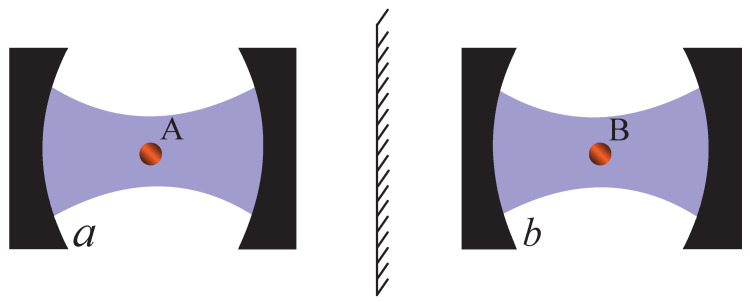
(Color online) In the setup, qubits *A* and *B* couple to the fields *a* and *b*, respectively. There is not any interaction between *A* and *B* or between *a* and *b*.

**Figure 2 entropy-21-00917-f002:**
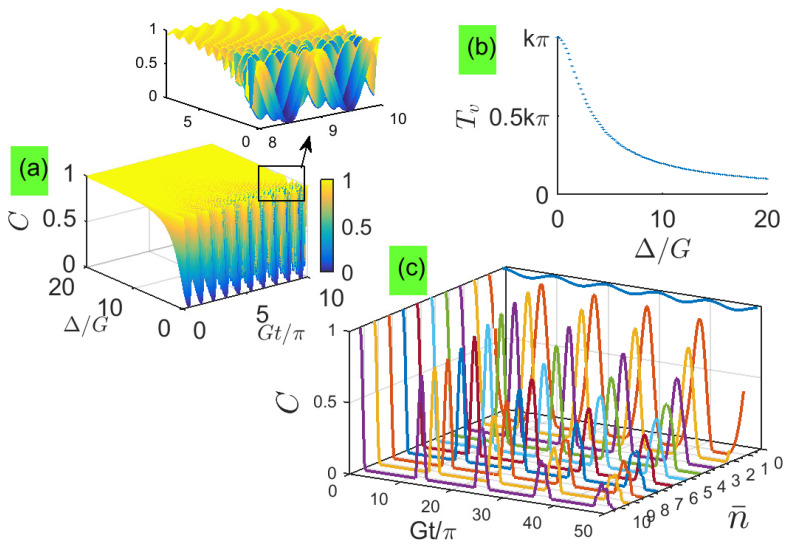
(Color online) For different detunings: (**a**) Time dependence of concurrence. (**b**) Period of concurrence. (**c**) The dynamics of the concurrence for increasing the average photon number $\bar{n}$, where $\bar{n}$ is starting from 0 (vacuum). To see periodic oscillations in Figure 2a clearly, a local-zoom subfigure is inserted in its top right-hand corner.

**Figure 3 entropy-21-00917-f003:**
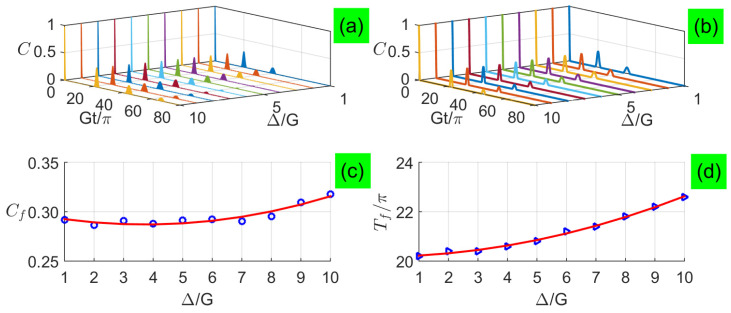
(Color online) Concurrence as a function of the evolution time under small detunings with $\bar{n}=100$: (**a**) analytical results and (**b**) numerical results, where analytical results are plotted based on Equation (Equation 37). Characteristics of the first revival envelope versus small detunings: (**c**) peaks $C_{f}$ and (**d**) periods $T_{f}$.

**Figure 4 entropy-21-00917-f004:**
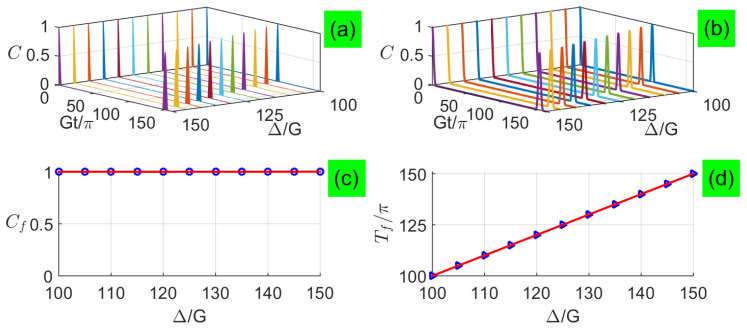
(Color online) Concurrence as a function of the evolution time under large detunings with $\bar{n}=100$: (**a**) analytical results and (**b**) numerical results, where analytical results are plotted based on Equation (Equation 43). Characteristics of the first revival envelope versus large detunings: (**c**) peaks $C_{f}$ and (**d**) periods $T_{f}$.

**Figure 5 entropy-21-00917-f005:**
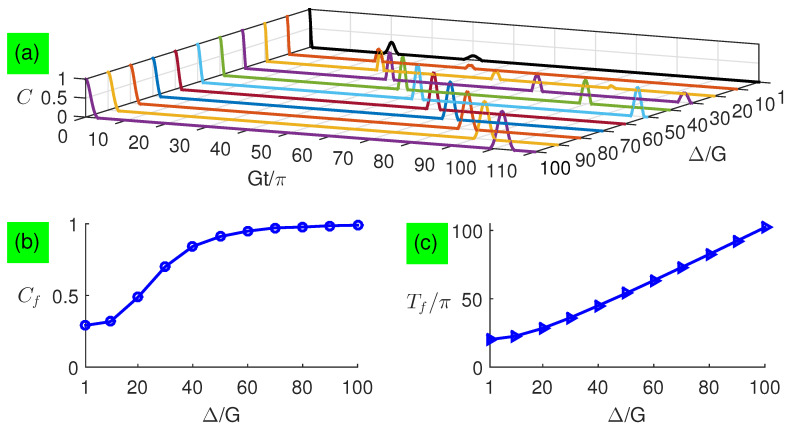
(Color online) (**a**) Concurrence as a function of the evolution time for general detunings with $\bar{n}=100$. Characteristics of the first revival envelope versus moderate detunings: (**b**) peaks $C_{f}$ and (**c**) periods $T_{f}$.

**Figure 6 entropy-21-00917-f006:**
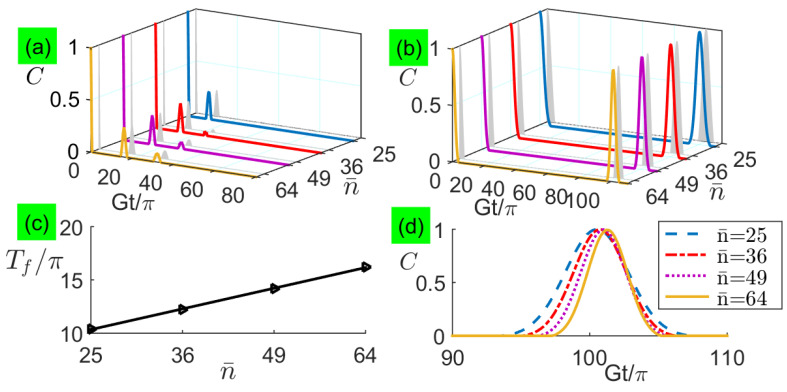
(Color online) Concurrence as a function of the evolution time for average photon number $\bar{n}$ with: (**a**) Δ=G and (**b**) Δ=100G. In each pair of curves with the same $\bar{n}$, the numerical result is plotted in front and the analytical result is plotted behind, where the analytical results in subfigures (**a**) and (**b**) are based on Equations (Equation 37) and (Equation 43) respectively. (**c**) Period $T_{f}$ of the first revival envelope versus average photon number $\bar{n}$ for Δ=G. (**d**) Concurrence dynamics of the first revival envelope versus average photon number $\bar{n}$ for Δ=100G.

**Figure 7 entropy-21-00917-f007:**
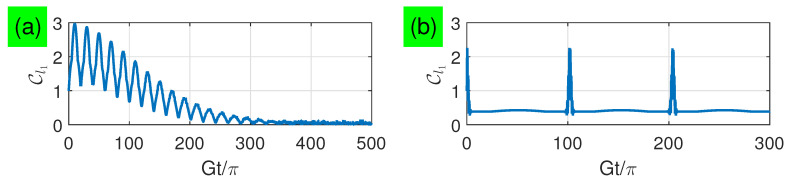
(Color online) $C_{l_{1}}$ as a function of the evolution time with $\bar{n}=100$ for: (**a**) Δ=G and (**b**) Δ=100G.

**Figure 8 entropy-21-00917-f008:**
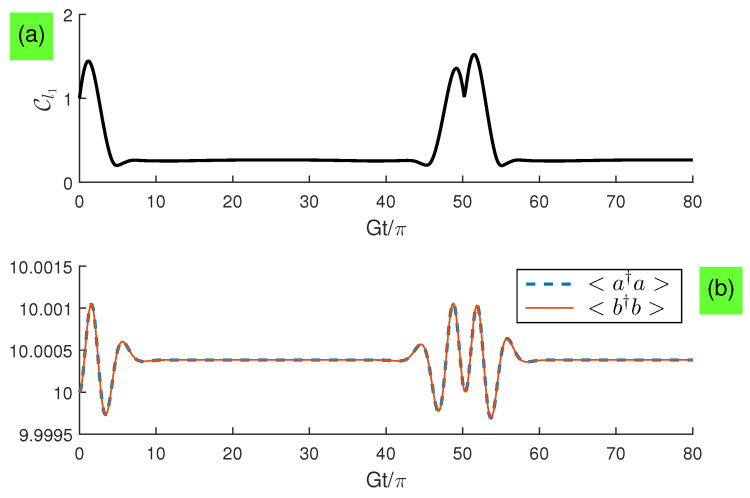
(Color online) Evolution dynamics with $\bar{n}=10$ and Δ=50G for: (**a**) $C_{l_{1}}$; (**b**) $<a^{†}a>$ and $<b^{†}b>$.

**Figure 9 entropy-21-00917-f009:**
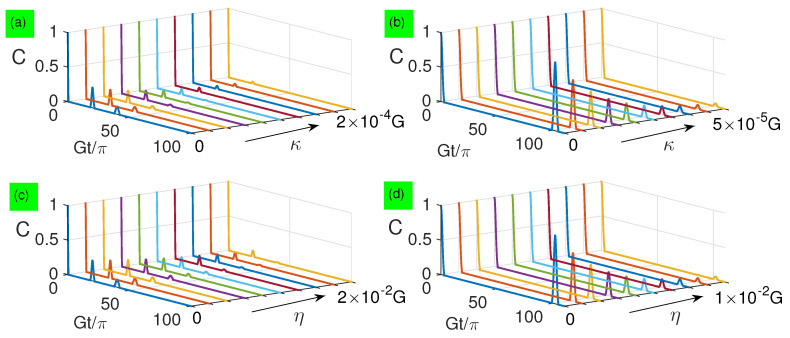
(Color online) Numerical simulation of entanglement dynamics based on Equation (Equation 46) when $\bar{n}=100$: (**a**) Δ=G and η=0; (**b**) Δ=100G and η=0; (**c**) Δ=G and κ=0; (**d**) Δ=100G and κ=0. Black arrows point to the direction of increasing dissipation.

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
