# Peer review of "Coherent State Control to Recover Quantum Entanglement and Coherence"

_entropy, 2019, doi:10.3390/e21100917_

Round 1

Reviewer 1 Report

The authors propose a generalization of known methods for studying the dynamics of entanglement and of coherence between two qubits embedded in separated cavities, under Jaynes-Cummings interactions (closed system without dissipation). While previous works fails to provide a solution when the qubit-field interaction is out-of-resonance, the authors here focus on generalizing the saddle point method (see Ref. [17]) to the case with small or large qubit-field detuning. This approach allow them to obtain two new analytical formulas for time-dependent concurrence, well describing the two-qubit entanglement dynamics. This result allows them to study how dynamical features of entanglement and coherence depend on the amount of qubit-cavity field detuning. In the final part of the manuscript, the authors move out from the closed system and consider the effects of dissipation on the two-qubit entanglement evolution, numerically simulating the system’s master equation by the approach of quantum trajectory.

I find the results of a certain interest in the field of open quantum systems and quantum dynamics. The manuscript has the merit to show new results in a more general situation of practical interest, namely a non-resonant interaction between the local qubit and its cavity field. The fact that the manuscript also treats the case of dissipation constitutes a merit of the work. However, the manuscript requires a revision to improve the overall presentation, especially concerning a poor account to the literature on the topics and a clear introductory motivation. I list in the following my comments and suggestions.

1) Page 1, Abstract, lines 8-9. The sentence should be corrected as: "[...], the maximal entanglement and coherence peaks can be fully and periodically retrieved, [...]". The same should be done in the main text (see line 48).

2) Page 1, Introduction, line 14. The first sentence of the paper introduces the main topic about entanglement and coherence preservation for qubits  and finally makes reference to Refs. [1-4]. Such a brief reference list appears quite poor and not up-to-date with respect to the state-of-art. To help the authors to retrieve a suitable literature for these important subjects, I suggest them to take into account at least all the following papers:

Rev. Mod. Phys. 86, 419 (2014).

Phys. Rev. Lett. 120, 240403 (2018).

Nature Comm. 8, 915 (2017).

Nature Comm. 4, 2851 (2013).

arXiv:1907.00136 [quant-ph].

Science 316, 555 (2007).

Optics Commun. 424, 26 (2018)

Laser Phys. Lett. 14, 055201 (2017).

Quantum Inf. Comput. 7, 1 (2007).

Rep. Prog. Phys. 78, 042001 (2015).

Quantum Inform. Process. 16, 191 (2017).

Int. J. Quantum Inform. 14, 1650031 (2016).

J. Opt. Soc. Am. B 36, 1858-1866 (2019).

Phys. Rev. Lett. 104, 250401 (2010).

Sci. Rep. 8, 14304 (2018)

Overview on the phenomenon of two-qubit entanglement revivals in classical environments, pages 367-391, DOI: https://doi.org/10.1007/978-3-319-53412-1_17 (2017). In: D. Soares-Pinto, F. Fanchini and G. Adesso (eds). "Lectures on general quantum correlations and their applications". Quantum Science and Technology. Springer, Cham.

Phys. Rev. Lett. 117, 160402 (2016).

Phys. Rev. Lett. 114, 210401 (2015).

Phys. Rev. A 84, 020304(R) (2011).

Phys. Rev. Lett. 107, 020502 (2011).

3) Page 1, line 19. At the end of the last sentence of the first paragraph, the authors should add Ref. [9] to Refs. [7, 8], due to the fact that Ref. [9] is just a review dedicated to entanglement sudden death, and also add the original paper: Phys. Rev. Lett. 93, 140404 (2004).

4) Page 1, line 20. After "Previous study" the authors should add some other review about the revivals of entanglement for two qubits in separated cavities, in particular:

Rep. Prog. Phys. 78, 042001 (2015).

Notice that these reference have been already suggested in point (1) above. Besides this, I would suggest the authors to clarify the meaning of the word "node" in this context, since it is not clear. For istance, they should explain if the nodes represent the quantum subsystems of interests, for instance qubits or two-level emitters. Or the nodes may be the local quantum field states to be entangled. Furthermore, the so-called nodes should be initially entangled in order to observe death and rebirth (revivals) of entanglement. Are the authors interested in entanglement and coherence between qubits (discrete variables) in cavity fields or are they interested in entanglement between cavity fields, for instance described by continuous variables like quadratures?

It is clear from section 1 and the model that they shall focus on qubit entanglement. However, the authors should better clarify these aspects in the introduction, in view of the motivation of the study.

5) Page 3, Eq. (7). In the expression of Eq. (7) the authors adopt the notation that the field states are written at the end of the ket. However, in calculating the (independent) evolutions, an initial subsystem qubit-field state has to be considered (see Eq. (8)). Therefore, the authors shoild explicitly write which field state is associated to each qubit (for example, the first qubit and the first field state in order of listing). The same holds for Eq. (14).

6) Page 17, Conclusions, line 307. As a notice of style, I would suggest the authors to write "The work is important because [...]", avoiding the term "fundamentally" which is typically associated to studies of fundamental quantum mechanics and appears to give too much emphasis. 

In conclusion, I believe the manuscript can be considered for publication provided that all the points above are satisfactorily fulfilled. 

Author Response

Dear Referee 1,

Thanks a lot for your positive attitude on our manuscript.

The followings are the detailed changes we have made:

1) Page 1, Abstract, lines 8-9. The sentence should be corrected as: "[...], the maximal entanglement and coherence peaks can be fully and periodically retrieved, [...]". The same should be done in the main text (see line 48).

We explain that: We have corrected this sentence as "[...], the maximal entanglement and coherence peaks can be fully and periodically retrieved, [...]" in the Abstract (see lines 8-9), the main text (see line 48), the conclusion (see line 304).

2) Page 1, Introduction, line 14. The first sentence of the paper introduces the main topic about entanglement and coherence preservation for qubits and finally makes reference to Refs. [1-4]. Such a brief reference list appears quite poor and not up-to-date with respect to the state-of-art. To help the authors to retrieve a suitable literature for these important subjects, I suggest them to take into account at least all the following papers: Rev. Mod. Phys. 86, 419 (2014);Phys. Rev. Lett. 120, 240403 (2018);Nature Comm. 8, 915 (2017);Nature Comm. 4, 2851 (2013);arXiv:1907.00136 [quant-ph];Science 316, 555 (2007);Science 323, 598 (2009);Optics Commun. 424, 26 (2018);Laser Phys. Lett. 14, 055201 (2017);Quantum Inf. Comput. 7, 1 (2007);Rep. Prog. Phys. 78, 042001 (2015);Quantum Inform. Process. 16, 191 (2017);Int. J. Quantum Inform. 14, 1650031 (2016);J. Opt. Soc. Am. B 36, 1858-1866 (2019);Phys. Rev. Lett. 104, 250401 (2010);Sci. Rep. 8, 14304 (2018);Overview on the phenomenon of two-qubit entanglement revivals in classical environments, pages 367-391, DOI: https://doi.org/10.1007/978-3-319-53412-1-17 (2017); In: D. Soares-Pinto, F. Fanchini and G. Adesso (eds). "Lectures on general quantum correlations and their applications". Quantum Science and Technology. Springer, Cham;Phys. Rev. Lett. 117, 160402 (2016);Phys. Rev. Lett. 114, 210401 (2015);Phys. Rev. A 84, 020304(R) (2011);Phys. Rev. Lett. 107, 020502 (2011).

We explain that: We have added these above papers as new references in the Introduction (see line 14).

3) Page 1, line 19. At the end of the last sentence of the first paragraph, the authors should add Ref. [9] to Refs. [7, 8], due to the fact that Ref. [9] is just a review dedicated to entanglement sudden death, and also add the original paper: Phys. Rev. Lett. 93, 140404 (2004).

We explain that: We have added the original Ref. [9] and the paper of Phys. Rev. Lett. 93, 140404 (2004) at the end of the last sentence of the first paragraph.

4) Page 1, line 20. After "Previous study" the authors should add some other review about the revivals of entanglement for two qubits in separated cavities, in particular:
Rep. Prog. Phys. 78, 042001 (2015).
Notice that these reference have been already suggested in point (1) above. Besides this, I would suggest the authors to clarify the meaning of the word "node" in this context, since it is not clear. For istance, they should explain if the nodes represent the quantum subsystems of interests, for instance qubits or two-level emitters. Or the nodes may be the local quantum field states to be entangled. Furthermore, the so-called nodes should be initially entangled in order to observe death and rebirth (revivals) of entanglement. Are the authors interested in entanglement and coherence between qubits (discrete variables) in cavity fields or are they interested in entanglement between cavity fields, for instance described by continuous variables like quadratures?
It is clear from section 1 and the model that they shall focus on qubit entanglement. However, the authors should better clarify these aspects in the introduction, in view of the motivation of the study.

We explain that: We have added the paper of "Rep. Prog. Phys. 78, 042001 (2015)" and some other review papers after "Previous study" in line 20. We clarify the meaning of the word "node" as: " The node represents the quantum subsystem of interest, i.e. a qubit and a local quantum field. Furthermore, the so-called nodes should be initially entangled in order to observe death and rebirth (revivals) of entanglement. We focus on qubit entanglement in this paper. ", and these sentences have been added after the sentence "Previous study [9] has shown ... decoherence in real experiments." in the second paragraph of Introduction.

5) Page 3, Eq. (7). In the expression of Eq. (7) the authors adopt the notation that the field states are written at the end of the ket. However, in calculating the (independent) evolutions, an initial subsystem qubit-field state has to be considered (see Eq. (8)). Therefore, the authors shoild explicitly write which field state is associated to each qubit (for example, the first qubit and the first field state in order of listing). The same holds for Eq. (14).

We explain that: We have added the new sentence "where the first qubit and the first field state are in order of listing " after Eq. (7) and Eq. (14), respectively.

6) Page 17, Conclusions, line 307. As a notice of style, I would suggest the authors to write "The work is important because [...]", avoiding the term "fundamentally" which is typically associated to studies of fundamental quantum mechanics and appears to give too much emphasis.

We explain that: We have deleted the term "fundamentally" in line 307 of Conclusion.

Best wishes,

Lituo Shen, Zhicheng Shi, Zhenbiao Yang

Reviewer 2 Report

In the manuscript entitled "Coherent-state control to recover quantum entanglement and coherence" the authors analyze the dynamics of the entanglement and quantum coherence of two qubits, each of them interacting separately with a cavity field mode. The interaction between the qubits and their corresponding field mode is considered to be of a Jaynes-Cummings form. This problem has been studied and addressed previously in a number of works (see for example Phys. Rev. Lett. 96, 080501 (2006) and Phys. Rev. A 82, 022321 (2010)); and it is of relevance to understand the dynamics of quantum correlations upon two-qubit entangled states are prepared, and it may be of interest to researchers working in the field of quantum entanglement and quantum optics.

The authors claim that they have extended previous results (see Ref. [17]) to account for a non-resonant interaction between qubit and field mode, which allows them to study an interesting case of a large detuning. The results are completed with a brief discussion about quantum coherence and the effect of dissipation. Although the results contained in the manuscript might deserve publication, it requires a major revision (see comments below). I thus cannot recommend the current version for publication. 

General comments:

- The first part of the manuscript is almost a one-to-one copy of Ref. [17] which largely diminishes the value of the subsequent analysis. I strongly recommend the authors to amend the first part; avoid the  discussions/expressions/materials taken from Ref. [17], and extend the original materials part: discussions on large detuning limit, quantum coherence and the effect of dissipation.

- The authors seem to overlook that Ref. [17] does consider a non-resonant interaction (see Sec. IV in [17]). For further developments however, a resonant condition was taken into account to further simplify the resulting equations. As the author generalize the results of this work, they should properly compare their expressions to those of Ref. [17].

- Since the authors want to pursue a generalization of this framework, and as the Jaynes-Cummings model is exactly solvable in any parameter regime, could they consider a case of distinct coupling constants (or detunings) for different cavities? What would be the impact of such an inhomogeneous interaction?

- In the large-detuning limit considered in Sec. 2.1, the Jaynes-Cummings interaction produces a vanishing small alteration onto the initial state as the field is initialized in a single Fock state. Thus, after a time t in this case, the evolved state is approximately equal to the initial one. Not surprisingly, as plotted in Fig. 2(a), the concurrence remains at 1 for large \Delta/G values simply because the stated started with C=1. As the authors show, this scenario will be different when considering highly-excited coherent states due to the relevance of the Stark shift. I recommend the authors to comment on this further and to provide examples in which the effect of the Stark shift can be clearly identified (e.g. the dynamics of the concurrence for increasing \bar{n}, starting from 0 (vacuum)).

Further issues:

1) What do the authors actually mean with "The X-state simplification, Fock-state shortcut and detuning-limit approximation work together in an amazingly accurate way" in the abstract and conclusions? Do they refer that the approximations agree with numerical results? 

2) The authors seem to suggest that Eq. (6) cannot be applied or "fails" when the detuning \Delta is different from zero. The concurrence given in Eq. (5) quantifies the entanglement, and holds for any two-qubit state. Please correct this.

3) The expressions given in Eqs. (20)-(34) generalize the ones given in Ref. [17] where \Delta=0 was assumed. The authors should comment or compare their resulting expressions when \Delta=0 with those obtained in Ref. [17]. This is briefly commented in Sec. 2.2.1 (line 191-192) but it should be extended (pointing to specific equations in Ref. [17]).

4) A comparison between Eq. (44) of the manuscript setting \Delta=0 (resonant case) and that of Ref. [17] (Eq. (60)) reveals mismatches. That is, Eq. (44) reduces to 1/4 (e^{-\tau^2/16\alpha^4}-1)+1/2 (e^{-\tau^2}\cos(4\alpha \tau))+\sum_{k=1,2..}1/(2\pi k) e^{-2(\tau-2\pi k\alpha)^2/(1+\pi^2k^2)}cos(4\alpha(\tau-\pi k \alpha)) while Eq. (60) in Ref. [17] reads as 1/4 (e^{-\tau^2/16\alpha^4}-1)+1/4 (e^{-\tau^2/2}\cos(4\alpha \tau))+\sum_{k=1,2..}1/(2\pi k) e^{-2(\tau-2\pi k\alpha)^2/(1+\pi^2k^2)}cos(4\alpha(\tau-2\pi k \alpha)). Please revise.

5) The plots in Fig. 3 show the small-detuning case. Is there any particular reason why the authors did not not the interesting case of a resonant interaction (\Delta=0)? 

6) How were the numerical simulations performed to obtained the numerical results presented in Secs. 2.2.1, 2.2.2, 2.2.3 and 2.3. Did the authors used the Hamiltonian given in Eq. (2) truncating the Fock basis? If so, what was this truncation? how did the authors simulate the dynamics of coherent states containing n=100 bosons in each of the cavity modes? Details must be provided.

Author Response

Dear Referee 2,

Thanks a lot for your positive attitude on our manuscript.

The followings are the detailed changes we have made:

1)The first part of the manuscript is almost a one-to-one copy of Ref. [17] which largely diminishes the value of the subsequent analysis. I strongly recommend the authors to amend the first part; avoid the discussions/expressions/materials taken from Ref. [17], and extend the original materials part: discussions on large detuning limit, quantum coherence and the effect of dissipation.

We explain that: The paragraph of "Since there is ... two-qubit concurrence [17] fails. " has been replaced by the sentence "We adopt Wootters concurrence C [39] as the two-qubit entanglement measure."

The paragraph of "When the fields are initially ... where $\rho_{22}$ $=\rho_{23}$ $=\rho_{32}$ $=\rho_{33}$ $=\frac{1}{2}$ $[\cos^{2}(\gamma_{1}t)$ $+\frac{\Delta^{2}}{4\gamma_{1}^{2}}\sin^{2}(\gamma_{1}t)]$,
and $\rho_{44}=\frac{G^{2}}{\gamma_{1}^{2}}\sin^{2}(\gamma_{1}t)$" has been replaced by the
sentence "When the fields are initially in their vacuum states,".

The sentences "Fig. 2(a) shows that ... without exhibiting sudden death." have been replaced by "Fig. 2(a) shows that the two-qubit entanglement keeps close to the maximum value $C=1$ for large detunings."

New sentences "For the two-qubit coherence, its analytical result is the same to concurrence, and the effect of dissipation will exponentially reduce the two-qubit entanglement, i.e. when $\alpha=0$ as proved in Eq. (47) later" have been added before the sentence "In the followings, we focus on the system ... with large amplitudes."

2)The authors seem to overlook that Ref. [17] does consider a non-resonant interaction (see Sec. IV in [17]). For further developments however, a resonant condition was taken into account to further simplify the resulting equations. As the author generalize the results of this work, they should properly compare their expressions to those of Ref. [17].

We explain that: New sentence "Compared with Eq. (27) of Ref. [38], the oscillation period of the expression $C$ here has an approximately inverse relation with $\Delta$ and becomes much smaller than that in Eq. (27) of Ref. [38]." has been added before the sentence "Fig. 2(a) shows that the two-qubit entanglement ... for large detunings."

3)Since the authors want to pursue a generalization of this framework, and as the Jaynes-Cummings model is exactly solvable in any parameter regime, could they consider a case of distinct coupling constants (or detunings) for different cavities? What would be the impact of such an inhomogeneous interaction?

We explain that: The case of distinct coupling constants for different cavities has been done in the new cited paper: "Dynamics of entanglement in Jaynes-Cummings nodes with nonidentical qubit-field coupling strengths, Entropy, 19, 331 (2017)", and the case of distinct detunings has a more complex calculation process which will be studied by us in the future. The impact of such an inhomogeneous interaction is that the qubit entanglement decays exponentially as the evolution time increases, exhibiting sudden death phenomenon, and the increasing gap accelerates the revival period and amplitude decay of the entanglement.

In the final paragraph of Appendix, we add a new sentence "Note that the case of distinct coupling constants for different cavities has been done in Ref. [45], but the case of distinct detunings for different cavities is still an open question."

4)In the large-detuning limit considered in Sec. 2.1, the Jaynes-Cummings interaction produces a vanishing small alteration onto the initial state as the field is initialized in a single Fock state. Thus, after a time t in this case, the evolved state is approximately equal to the initial one. Not surprisingly, as plotted in Fig. 2(a), the concurrence remains at 1 for large $\Delta/G$ values simply because the stated started with C=1. As the authors show, this scenario will be different when considering highly-excited coherent states due to the relevance of the Stark shift. I recommend the authors to comment on this further and to provide examples in which the effect of the Stark shift can be clearly identified (e.g. the dynamics of the concurrence for increasing $\bar{n}$, starting from 0 (vacuum)).

We explain that: We add a new figure [see Fig. 2(c)] and caption to provide examples in which the effect of the Stark shift can be clearly identified (e.g. the dynamics of the concurrence for increasing $\bar{n}$, starting from 0 (vacuum)), as shown in Fig. 2(c).

New paragraph "To provide examples in which the effect of the Stark shift can be clearly identified, the dynamics of the concurrence for increasing the average photon number $\bar{n}$ is plotted in Fig. 2(c), where $\bar{n}$ is starting from 0 (vacuum). When $\bar{n}>0$, the Stark shift can be clearly identified that the concurrence can not remain at 1 and becomes smaller as the evolution time is longer." has been added before the paragraph "To explain the above results, ... as proved in Eq. (47) later."

Further issues:

1) What do the authors actually mean with "The X-state simplification, Fock-state shortcut and detuning-limit approximation work together in an amazingly accurate way" in the abstract and conclusions? Do they refer that the approximations agree with numerical results?

We mean that the analytical approximations agree with numerical results.

The sentence "The X-state simplification, Fock-state shortcut and detuning-limit
approximation work together in an amazingly accurate way." in abstract and conclusion has been
changed as "The X-state simplification, Fock-state shortcut and detuning-limit
approximation work together in an amazingly accurate way, which agrees with numerical results."

2) The authors seem to suggest that Eq. (6) cannot be applied or "fails" when the detuning $\Delta$ is different from zero. The concurrence given in Eq. (5) quantifies the entanglement, and holds for any two-qubit state. Please correct this.

We explain that: The sentence "When there is detuning, the previous analytic formula for two-qubit concurrence [17] fails." has been deleted.

3) The expressions given in Eqs. (20)-(34) generalize the ones given in Ref. [17] where $\Delta=0$ was assumed. The authors should comment or compare their resulting expressions when $\Delta=0$ with those obtained in Ref. [17]. This is briefly commented in Sec. 2.2.1 (line 191-192) but it should be extended (pointing to specific equations in Ref. [17]).

We explain that: The sentence "Compared with Eq. (60) of Ref. [38], this formula here in Eq. (37) contains new contributions of $\Delta^2$ in the index of exponential function
and the period of cosine function." has been added after the sentence "For this formula we ... main contribution to the sums".

The sentence "Compared with Eq. (61) of Ref. [38], this formula here in Eq. (39) contains a new contribution $2\Delta^2\tau^2/(64\alpha^2G^2)$ in the index of exponential function, and the relative revival envelope height grows with the increasing $\Delta$." has been added before the sentence "Thus the formula in Eq. (37) is ... a special limit of small detuning".

4) A comparison between Eq. (44) of the manuscript setting $\Delta=0$ (resonant case) and that of Ref. [17] (Eq. (60)) reveals mismatches. That is, Eq. (44) reduces to $1/4 (e^{-\tau^2/16\alpha^4}-1)+1/2 (e^{-\tau^2}\cos(4\alpha \tau))+\sum_{k=1,2..}1/(2\pi k) e^{-2(\tau-2\pi k\alpha)^2/(1+\pi^2k^2)}cos(4\alpha(\tau-\pi k \alpha))$ while Eq. (60) in Ref. [17] reads as $1/4 (e^{-\tau^2/16\alpha^4}-1)+1/4 (e^{-\tau^2/2}\cos(4\alpha \tau))+\sum_{k=1,2..}1/(2\pi k) e^{-2(\tau-2\pi k\alpha)^2/(1+\pi^2k^2)}cos(4\alpha(\tau-2\pi k \alpha)).$ Please revise.

We explain that: This is a typing mistake, so we have revised the corresponding equations, as shown in the new Eq. (37).

5) The plots in Fig. 3 show the small-detuning case. Is there any particular reason why the authors did not plot the interesting case of a resonant interaction ($\Delta=0$)?

We explain that: The interesting case of a resonant interaction ($\Delta=0$) has been plotted in the Fig. 5 of Ref. [38](i.e. PRA 82, 022321, 2010), to avoid a duplication with the figure of Ref. [38], we did not plot the interesting case of a resonant interaction ($\Delta=0$).

6) How were the numerical simulations performed to obtained the numerical results presented in Secs. 2.2.1, 2.2.2, 2.2.3 and 2.3. Did the authors used the Hamiltonian given in Eq. (2) truncating the Fock basis? If so, what was this truncation? how did the authors simulate the dynamics of coherent states containing n=100 bosons in each of the cavity modes? Details must be provided.

We explain that: To perform the numerical simulations, we used the Hamiltonian given in Eq. (2) by truncating the Fock basis and calculating the reduced density matrix
in each excitation subspace, then summing over all the subspaces
to get the total reduced density matrix. This Fock-basis truncation is based on the precision accuracy of numerical concurrence, and the total excitation number is cut off at $2\bar{n}$.
For example, when coherent states contain $\bar{n}=100$ bosons in each of the cavity modes, the total excitation number is cut off at $200$. This is because when $\bar{n}>>1$, the width of the photon number distribution obeys $1<<\Delta n<<\bar{n}$. Therefore, it is safe to make this Fock-basis truncation.

These explanations have been added before the sentence "In Figs. 3(a) and 3(b), we plot the long-time entanglement dynamics ..., respectively."

Best wishes,

Lituo Shen, Zhicheng Shi, Zhenbiao Yang

Reviewer 3 Report

This paper studies entanglement dynamics of a two-qubit system, where each qubit interacts with a single mode cavity photon. The system is described by the two independent Jaynes-Cumming models with finite qubit-photon detuning. The authors investigate how the detuning affects the dynamics of two-qubit entanglement. They provide thorough calculation including small and large detuning limits. I think that the results given in this manuscript is sound but the manuscript is luck of physical explanation for the results.  The authors should explain not only physical meanings and importance of the results. Furthermore I have some comments.   (1) Usually the Hamiltonian in the interaction picture is defined by $$e^{iH_{0}t/\hbar}H_{I}e^{-iH_{0}t/\hbar}$$. The authors should explain what $$H_{0}$$ and $$H_{I}$$ are in deriving Equation (2) from Equation (1).   (2) The authors should provide discussions on the effects of the detuning from physical point of view. They write, for instance, on Line 92, “These behaviors are very different from the system with resonant couplings.”  Why does detuning make such a difference? Physical explanation is needed. Similar comment is applied to the other parts of the manuscript.   (3) Taking account of dissipation is very important. So the authors should explain the model in detail. In the present manuscript, it is not clear whether Equation (53) is an interaction Hamiltonian in the Schrodinger picture or in the interaction picture. Furthermore they should provide a outline of numerical calculation.   In conclusion, I cannot recommend publication in the present form of the manuscript.

Author Response

Dear Referee 3,

Thanks a lot for your positive attitude on our manuscript.

The followings are the detailed changes we have made:

I think that the results given in this manuscript is sound but the manuscript is luck of physical explanation for the results. The authors should explain not only physical meanings and importance of the results.

We have added more physical explanations for the results [see our reply to the comment (2)].

(1) Usually the Hamiltonian in the interaction picture is defined by $$e^{iH_{0}t/\hbar}H_{I}e^{-iH_{0}t/\hbar}$$. The authors should explain what $$H_{0}$$ and $$H_{I}$$ are in deriving Equation (2) from Equation (1).

We explain that: in deriving Equation (2) from Equation (1), $H_{0}=\frac{\omega}{2}(\sigma_{z}^{A}+\sigma_{z}^{B})+\omega a^{\dagger}a+\omega b^{\dagger}b$,
and $H_{I}=H-H_{0}$, which is the same with the previous Eq. (2).

New sentence "where $H_{0}=\frac{\omega}{2}(\sigma_{z}^{A}+\sigma_{z}^{B})+\omega a^{\dagger}a+\omega b^{\dagger}b$ and $H_{I}=e^{iH_{0}t/\hbar}(H-H_{0})e^{-iH_{0}t/\hbar}$ ($\hbar=1$)." has been added after the Eq. (2).

(2) The authors should provide discussions on the effects of the detuning from physical point of view. They write, for instance, on Line 92, "These behaviors are very different from the system with resonant couplings." Why does detuning make such a difference? Physical explanation is needed. Similar comment is applied to the other parts of the manuscript.

We explain that: This is because detuning reduces the energy-exchange probability between the qubit and its local photon field, changing the period and amplitude of the Rabi oscillation. As the detuning increases, the energy coupling between entangled qubits and their respective vacuum states is very week, and the excitation energy mainly keeps in two qubits as the evolution time increases, making two qubits be always in the originally entangled state.

These explanations have been added after the sentence "These behaviors are very ... oscillation with a fixed period."

For the other parts of the manuscript (i.e., 2.2. Coherent-state fields), we explain that: This is because the entanglement between the two subsystems is conserved due to no interaction between them. The small detuning in each subsystem makes each photon field deviate from the coherent state so that an ebit for the
qubits cannot be completely transferred to the fields, i.e., two
qubits become a mixed state and their entanglement quickly vanishes by tracing out two fields as the evolution time increases, so that the two qubits can not be fully recovered.

These explanations have been added after the sentence "The numerical results in Fig. 3(b) are
not perfectly predicted by ... in the new formula of Eq. (37)."

(3) Taking account of dissipation is very important. So the authors should explain the model in detail. In the present manuscript, it is not clear whether Equation (53) is an interaction Hamiltonian in the Schrodinger picture or in the interaction picture.

We explain that: Equation (53) is an interaction Hamiltonian in the Schr\"{o}dinger picture, where the operator is time-independent but the state vector is time-dependent.

These explanations have been added in after the sentence "where ... operators causing
instantaneous quantum jumps."

(4) Furthermore they should provide a outline of numerical calculation.

We explain that: To perform the numerical simulations, we used the Hamiltonian given in Eq. (2) by truncating the Fock basis and calculating the reduced density matrix
in each excitation subspace, then summing over all the subspaces
to get the total reduced density matrix. This Fock-basis truncation is based on the precision accuracy of numerical concurrence, and the total excitation number is cut off at $2\bar{n}$.
For example, when coherent states contain $\bar{n}=100$ bosons in each of the cavity modes, the total excitation number is cut off at $200$. This is because when $\bar{n}>>1$, the width of the photon number distribution obeys $1<<\Delta n<<\bar{n}$. Therefore, it is safe to make this Fock-basis truncation.

These explanations have been added before the sentence "In Figs. 3(a) and 3(b), we plot the long-time entanglement dynamics ..., respectively."

Best wishes,

Lituo Shen, Zhicheng Shi, Zhenbiao Yang

Reviewer 4 Report

This paper is on the problem how the quantum entanglement between two qubits would evolve in time when each qubit individually interacts with identical cavity modes. Similar work has been done by M. Yonac and J. H. Eberly in 2010(Ref[17]) by considering the cases of cavity vacuum state and coherent states. The authors considered the same problem while included the finite detuning between the qubit energy and cavity photon energy. In principle, they used the same approximation that the photon number is large so a coherent state can be approximated using Poisson distribution, and a few mathematical approximations such that saddle point approximation is applied when integrating the photon number states. Some analytical formulas can be obtained in the small detuning and large detuning cases and they show good agreement with the numerical results. In general, they could observe the sudden death of concurrence and revival as previous works. For the small detuning case, the concurrence does not fully revive in a short time, while for the large detuning case, the revivals could be as large as 100%.

In summary, the authors did not present new physics in this work, or they did not emphasize if there is any. Nevertheless, they provided clear formulas for this problem and could be useful for future works. I personally feel that Sec. 2.3 and Sec. 3 could draw me more attention but unfortunately no much results and discussions on the two aspects. Anyway, this paper is publishable if the following questions can be answered.

-Eq. (3) and (4) describes the coherent oscillation between |e, n> and |g, n+1>, but many other quantum states with different photon numbers are truncated. Actually, these results would only be correct when the energies of the two states are close to each other and those of others are not. If it is not the case, additional quantum states could be involved in the oscillation and the simple results would breakdown. When the authors mentioned that their results can be used for large detuned cases, their assumptions have to be clarified first.

-According to Eq. (10), the concurrence is a periodic function of time and the entanglement revives after a period. However, Fig. 2(a) illustrates that some decay occurs in time. Is there any additional assumption in calculating the plot?

-In Fig 3(c), is Cf value for the first concurrence revival peak?
Since the concurrence gradually reduces as time goes by, which peak is selected should be noted clearly.

-In Fig. 3 and 4 caption, should "peek" be peak?

-It would be interesting to see what are the photon states when the qubit states evolve to a maximally coherence and minimally coherence. For example, if we consider the approximation of Poisson distribution, how the average photon number and width in distribution evolve in time?

Author Response

Dear Referee 4,

Thanks a lot for your positive attitude on our manuscript.

The followings are the detailed changes we have made:

1) Eq. (3) and (4) describes the coherent oscillation between $|e, n>$ and $|g, n+1>$, but many other quantum states with different photon numbers are truncated. Actually, these results would only be correct when the energies of the two states are close to each other and those of others are not. If it is not the case, additional quantum states could be involved in the oscillation and the simple results would breakdown. When the authors mentioned that their results can be used for large detuned cases, their assumptions have to be clarified first.

We explain that: new assumption sentences "Consider the coherent oscillation between $|e, n>$ and $|g, n+1>$, many other quantum states with different photon numbers are truncated in the following results. Actually, these results would only be correct when the energies of the two states are close to each other and those of others are not. If it is not the case, additional quantum states could be involved in the oscillation and the following simple results would break down." have been added after the sentence "Note that the qubit excitation ... the resonant Jaynes-Cummings interaction."

2) According to Eq. (10), the concurrence is a periodic function of time and the entanglement revives after a period. However, Fig. 2(a) illustrates that some decay occurs in time. Is there any additional assumption in calculating the plot?

We explain that: Fig. 2(a) is not easy to see its periodic oscillation clearly, and there is not decay occurs in time. The look-like gap (or "decay") in the figure is caused by the continuation change of periods with the detuning. We revised Fig. 2(a) by adding a local-zoom subfigure in its top right-hand corner.

3) In Fig 3(c), is Cf value for the first concurrence revival peak?
Since the concurrence gradually reduces as time goes by, which peak is selected should be noted clearly.

We explain that: In Fig. 3(c), $C_{f}$ value is the first concurrence revival peak.

New sentence "In Fig. 3(c), $C_{f}$ value is the first concurrence revival peak." is added before the sentence "We find that the entanglement ... as shown in Figs. 3(c)."

4) In Fig. 3 and 4 caption, should "peek" be peak?

We explain that: In Fig. 3 and 4 captions, the word "peek" has been changed to "peak".

5) It would be interesting to see what are the photon states when the qubit states evolve to a maximally coherence and minimally coherence. For example, if we consider the approximation of Poisson distribution, how the average photon number and width in distribution evolve in time?

We explain that: A new figure (see Fig. 8) has been added after Fig. 7. As shown in Fig. 8, we find that the average photon number for each subsystem becomes maximum when the qubit states evolve to a maximally coherence, but becomes minimum when the qubit states evolve to a minimally coherence, and width in distribution has the similar evolution with that in the qubit coherence.

New paragraph "It would be interesting to see what are the photon states when the qubit states evolve to a maximally coherence and minimally coherence. In Fig. 8, we plot the evolution dynamics of the average photon number and width in distribution. We find that the average photon number for each subsystem becomes maximum when the qubit states evolve to a maximally coherence, but becomes minimum when the qubit states evolve to a minimally coherence, and width in distribution has the similar evolution with that in the qubit coherence." have been added after the sentence "But, when the detuning is small, the coherence may inversely increase during the vanishing of entanglement."

Best wishes,

Lituo Shen, Zhicheng Shi, Zhenbiao Yang

Round 2

Reviewer 1 Report

The authors have fulfilled my previous comments and criticisms. The manuscript can be now accepted for publication.

Reviewer 2 Report

The authors have adequately addressed all my comments and amended the manuscript where necessary. I therefore recommend the revised manuscript for publication in Entropy.

Reviewer 3 Report

I think that the authors replied satisfactorily to all my comments on the original manuscript. Thus I recommend publication of the revised manuscript in Entropy.

Reviewer 4 Report

Overall I found that the authors' replies are acceptable, except some text error, such as On line 84, page 3, "...between entangled qubits and their respective vacuum states is very week,", "week" should be "weak". But the sentence needs revision because of grammar problems.

There are other sentences having similar language problems. The authors should have the manuscript grammar checked before publication.